# Bellman Residual Orthogonalization
# for Offline Reinforcement Learning

**Andrea Zanette**
Department of Computer Sciences and Electrical Engineering
University of California, Berkeley
`zanette@berkeley.edu`

**Martin J. Wainwright**
Department of Electrical Engineering and Computer Sciences,
Department of Mathematics,
Massachusetts Institute of Technology, and
Department of Computer Sciences and Electrical Engineering,
Department of Statistics,
University of California, Berkeley
`wainwrigwork@gmail.com`

## Abstract

We propose and analyze a reinforcement learning principle that approximates the Bellman equations by enforcing their validity only along an user-defined space of test functions. Focusing on applications to model-free offline RL with function approximation, we exploit this principle to derive confidence intervals for off-policy evaluation, as well as to optimize over policies within a prescribed policy class. We prove an oracle inequality on our policy optimization procedure in terms of a trade-off between the value and uncertainty of an arbitrary comparator policy. Different choices of test function spaces allow us to tackle different problems within a common framework. We characterize the loss of efficiency in moving from on-policy to off-policy data using our procedures, and establish connections to concentrability coefficients studied in past work. We examine in depth the implementation of our methods with linear function approximation, and provide theoretical guarantees with polynomial-time implementations even when Bellman closure does not hold.

## 1   Introduction

Markov decision processes (MDP) provide a general framework for optimal decision-making in sequential settings (e.g., [Put94, Ber95a, Ber95b]). Reinforcement learning refers to a general class of procedures for estimating near-optimal policies based on data from an unknown MDP (e.g., [BT96, SB18]). Different classes of problems can be distinguished depending on our access to the data-generating mechanism. Many modern applications of RL involve learning based on a pre-collected or offline dataset. Moreover, the state-action spaces are often sufficiently complex that it becomes necessary to implement function approximation. In this paper, we focus on model-free offline reinforcement learning (RL) with function approximation, where prior knowledge about the MDP is encoded via the value function. In this setting, we focus on two fundamental problems: (1) offline policy evaluation—namely, the task of accurately predicting the value of a target policy; and (2) offline policy optimization, which is the task of finding a high-performance policy.

36th Conference on Neural Information Processing Systems (NeurIPS 2022).

There are various broad classes of approaches to off-policy evaluation, including importance sampling [Pre00, TB16, JL16, LLTZ18], as well as regression-based methods [LP03, MS08, CJ19]. Many methods for offline policy optimization build on these techniques, with a line of recent papers including the addition of pessimism [JYW21, XCJ+21, ZWB21]. We provide a more detailed summary of the literature in Appendix A.3.

In contrast, this work investigates a different model-free principle—different from importance sampling or regression-based methods—to learn from an offline dataset. It belongs to the class of weight learning algorithms, which leverage an auxiliary function class to either encode the marginalized importance weights of the target policy [LLTZ18, XJ20b], or estimates of the Bellman errors [ASM08, CJ19, XJ20b]. Some work has considered kernel classes [FRTL20] or other weight classes to construct off-policy estimators [UHJ20] as well as confidence intervals at the population level [JH20]. However, these works do not examine in depth the statistical aspects of the problem, nor elaborate upon the design of the weight function classes.[1] The last two considerations are essential to obtaining data-dependent procedures accompanied by rigorous guarantees, and to provide guidance on the choice of weight class, which are key contributions of this paper.

For space reasons, we motivate our approach in the idealized case where the Bellman operator is known in Appendix A.1, and compare with the weight learning literature at the population level in Appendix A.2. Let us summarize our main contributions in the following three paragraphs.

**Conceptual contributions:** Our paper makes two novel contributions of conceptual nature:

1. We propose a method, based on *approximate empirical orthogonalization* of the Bellman residual along test functions, to construct confidence intervals and to perform policy optimization.

2. We propose a sample-based approximation of such principle, based on *self-normalization* and *regularization*, and obtain general guarantees for parametric as well as non-parametric problems.

The construction of the estimator, its statistical analysis, and the concrete consequences (described in the next paragraph) are the major distinctions with respect to past work on weight learning methods [UHJ20, JH20]. Our analysis highlights the statistical trade-offs in the choice of the test functions. (See Appendix A.2 for comparison with past work at the population level.)

**Domain-specific results:** In order to illustrate the broad effectiveness and applicability of our general method and analysis, we consider several domains of interest. We show how to recover various results from past work—and to obtain novel ones—by making appropriate choices of the test functions and invoking our main result. Among these consequences, we discuss the following:

1. When marginalized importance weights are available, they can be used as test class. In this case we recover a similar results as the paper [XJ20b]; however, here we only require concentrability with respect to a comparator policy instead of over all policies in the class.

2. When some knowledge of the Bellman error class is available, it can be used as test class. Similar results have appeared previously either with stronger concentrability [CJ19] or in the special case of Bellman closure [XCJ+21].

3. We provide a test class that projects the Bellman residual along the error space of the $\mathcal{Q}$ class. The resulting procedure is as an extension of the LSTD algorithm [BB96] to non-linear spaces, which makes it a natural approach if no domain-specific knowledge is available. A related result is the lower bound by [FKSLX21], which proves that without Bellman closure learning is hard even with small density ratios. In contrast, our work shows that learning is still possible even with large density ratios.

4. Finally, our procedure inherits some form of "multiple robustness". For example, the two test classes corresponding to Bellman completeness and marginalized importance weights can be used together, and guarantees will be obtained if *either* Bellman completeness holds or the importance weights are correct. We examine this issue in Section 4.4.

**Linear setting:** We examine in depth an application to the linear setting, where we propose the first *computationally tractable* policy optimization procedure *without assuming Bellman completeness*.

---

[1] For instance, the paper [FRTL20] only shows validity of ther intervals, not a performance bound; on the other hand, the paper [JH20] gives analyses at the population level, and so does not address the alignment of weight functions with respect to the dataset in the construction of the empirical estimator, which we do via self-normalization and regularization. This precludes obtaining the same type of guarantees that we present here.

The closest result here is given in the paper [ZWB21], which holds under Bellman closure. Our procedure can be thought of making use of LSTD-type estimates so as to establish confidence intervals for the projected Bellman equations, and then using an iterative scheme for policy improvement.

## 2 Background and set-up

We begin with some notation used throughout the paper. For a given probability distribution $\rho$ over a space $\mathcal{X}$, we define the $L^2(\rho)$-inner product and semi-norm as $\langle f_1, f_2 \rangle_\rho = \mathbb{E}_\rho[f_1 f_2]$, and $\|f_1\|_\rho = \sqrt{\langle f_1, f_1 \rangle_\rho}$. The identity function that returns one for every input is denoted by $\mathbb{1}$. We frequently use notation such as $c, c', \tilde{c}, c_1, c_2$ etc. to denote constants that can take on different values in different sections of the paper.

### 2.1 Markov decision processes and Bellman errors

We focus on infinite-horizon discounted Markov decision processes [Put94, BT96, SB18] with discount factor $\gamma \in [0, 1)$, state space $\mathcal{S}$, and an action set $\mathcal{A}$. For each state-action pair $(s, a)$, there is a reward distribution $R(s, a)$ supported in $[0, 1]$ with mean $r(s, a)$, and a transition $\mathbb{P}(\cdot \mid s, a)$.

A (stationary) stochastic policy $\pi$ maps states to actions. For a given policy, its $Q$-function is the discounted sum of future rewards based on starting from the pair $(s, a)$, and then following the policy $\pi$ in all future time steps $Q^\pi(s, a) = r(s, a) + \sum_{h=0}^\infty \gamma^h \mathbb{E}[r_h(S_h, A_h) \mid (S_0, A_0) = (s, a)]$, where the expectation is taken over trajectories with $A_h \sim \pi(\cdot \mid S_h)$, and $S_{h+1} \sim \mathbb{P}(\cdot \mid S_h, A_h)$ for $h = 1, 2, \ldots$. We also use $Q^\pi(s, \pi) = \mathbb{E}_{A \sim \pi(\cdot \mid s)} Q^\pi(s, A)$ and define the *Bellman evaluation operator* as $(\mathcal{T}^\pi Q)(s, a) = r(s, a) + \mathbb{E}_{S^+ \sim \mathbb{P}(\cdot \mid s, a)} Q(S^+, \pi)$. The value function satisfies $V^\pi(s) = Q^\pi(s, \pi)$. In our analysis, we assume that policies have action-value functions that satisfy the uniform bound $\sup_{(s,a)} |Q^\pi(s, a)| \leq 1$. We are also interested in approximating optimal policies, whose value and action-value functions are defined as $V^\star(s) = V^{\pi^\star}(s) = \sup_\pi V^\pi(s)$ and $Q^\star(s, a) = Q^{\pi^\star}(s, a) = \sup_\pi Q^\pi(s, a)$.

We assume that the starting state $S_0$ is drawn according to $\nu_{\text{start}}$ and study $V^\pi = \mathbb{E}_{S_0 \sim \nu_{\text{start}}}[V^\pi(S_0)]$. The *discounted occupancy measure* associated with a policy $\pi$ is defined as $d_\pi(s, a) = (1 - \gamma) \sum_{h=0}^\infty \gamma^h \mathbb{P}_h[(S_h, A_h) = (s, a)]$. We adopt the shorthand notation $\mathbb{E}_\pi$ for expectations over $d_\pi$. For any functions $f, g : \mathcal{S} \times \mathcal{A} \to \mathbb{R}$, we make frequent use of the shorthands $\mathbb{E}_\pi[f] \overset{def}{=} \mathbb{E}_{(S,A) \sim d_\pi}[f(S, A)]$, and $\langle f, g \rangle_\pi \overset{def}{=} \mathbb{E}_{(S,A) \sim d_\pi}[f(S, A) g(S, A)]$. Note moreover that we have $\langle \mathbb{1}, f \rangle_\pi = \mathbb{E}_\pi[f]$ where $\mathbb{1}$ denotes the identity function.

For a given $Q$-function and policy $\pi$, let us define the *temporal difference error* (or TD error) associated with the sample $z = (s, a, r, s^+)$ and the *Bellman error* at $(s, a)$

$$(\delta^\pi Q)(z) \overset{def}{=} Q(s, a) - r - \gamma Q(s^+, \pi), \qquad (\mathcal{B}^\pi Q)(s, a) \overset{def}{=} Q(s, a) - r(s, a) - \gamma \mathbb{E}_{s^+ \sim \mathbb{P}(s, a)} Q(s^+, \pi).$$

The TD error is a random variable function of $z$, while the Bellman error is its conditional expectation with respect to the immediate reward and successor state at $(s, a)$. Many of our bounds involve the quantity $\mathbb{E}_\pi \mathcal{B}^\pi Q = \mathbb{E}_{(S,A) \sim d_\pi}[\mathcal{B}^\pi Q(S, A)]$.

Finally, we introduce the data generation mechanism. A more general sampling model is described in Appendix B.

**Assumption 1** (I.i.d. dataset). *An i.i.d. dataset is a collection $\mathcal{D} = \{(s_i, a_i, r_i, s_i^+, o_i)\}_{i=1}^n$ such that for each $i = 1, \ldots, n$ we have $(s_i, a_i, o_i) \sim \mu$ and conditioned on $(s_i, a_i, o_i)$, we observe a noisy reward $r_i = r(s_i, a_i) + \eta_i$ with $\mathbb{E}[\eta_i \mid \mathcal{F}_i] = 0$, $|r_i| \leq 1$ and the next state $s_i^+ \sim \mathbb{P}(s_i, a_i)$.*

### 2.2 Function Spaces and Weak Representation

Our methods involve three different types of function spaces, corresponding to policies, action-value functions, and test functions. A test function $f$ is a mapping $(s, a, o) \mapsto f(s, a, o)$ such that $\sup_{(s,a,o)} |f(s, a, o)| \leq 1$, where $o$ is an optional identifier containing side information. Our methodology involves the following three function classes:

- a *policy class* $\Pi$ that contains all policies $\pi$ of interest (for evaluation or optimization);

- for each $\pi$, the *predictor class* $\mathcal{Q}^\pi$ of action-value functions $Q$ that we permit; and

- for each $\pi$, the *test function class* $\mathcal{F}^\pi$ that we use to enforce the Bellman residual constraints.

We use the shorthands $\mathcal{Q} = \cup_{\pi \in \Pi} \mathcal{Q}^\pi$ and $\mathcal{F} = \cup_{\pi \in \Pi} \mathcal{F}^\pi$. We assume *weak realizability*:

**Assumption 2** (Weak Realizability). *For a given policy $\pi$, the predictor class $\mathcal{Q}^\pi$ is weakly realizable with respect to the test space $\mathcal{F}^\pi$ and the measure $\mu$ if there exists a predictor $Q_\star^\pi \in \mathcal{Q}^\pi$ such that*

$$\langle f, \mathcal{B}^\pi Q_\star^\pi \rangle_\mu = 0 \text{ for all } f \in \mathcal{F}^\pi \qquad \text{and} \qquad \langle \mathbb{1}, \mathcal{B}^\pi Q_\star^\pi \rangle_\pi = 0. \tag{1}$$

The first condition requires the predictor to satisfy the Bellman equations *on average*. The second condition amounts to requiring that the predictor returns the value of $\pi$ at the start distribution: using Lemma 9 stated in the sequel, we have

$$\mathbb{E}_{S \sim \nu_{\text{start}}} Q_\star^\pi(S, \pi) - V^\pi = \mathbb{E}_{S \sim \nu_{\text{start}}}[Q_\star^\pi - Q^\pi](S, \pi) = \frac{1}{1-\gamma} \mathbb{E}_\pi \mathcal{B}^\pi Q_\star^\pi = \frac{1}{1-\gamma} \langle \mathbb{1}, \mathcal{B}^\pi Q_\star^\pi \rangle_\pi = 0.$$

This weak notion should be contrasted with *strong realizability*, which requires a function $Q^\pi \in \mathcal{Q}^\pi$ that satisfies the Bellman equation in all state-action pairs.

A stronger assumption that we sometime use is Bellman closure, which requires that $\mathcal{T}^\pi(Q) \in \mathcal{Q}^\pi$ for all $Q \in \mathcal{Q}^\pi$. The corresponding 'weak' version is given in Appendix A.4.

# 3 Policy Estimates via the Weak Bellman Equations

In this section, we introduce our high-level approach, first at the population level and then in terms of regularized/normalized sample-based approximations.

## 3.1 Weak Bellman equations, empirical approximations and confidence intervals

We begin by noting that the predictor $Q^\pi$ satisfies the Bellman equations everywhere in the state-action space, i.e., $\mathcal{B}^\pi Q^\pi = 0$. However, if our dataset is "small" relative to the complexity of (functions) on the state-action space, then it is unrealistic to enforce such a stringent condition. Instead, the idea is to control the Bellman error in a weighted-average sense, where the weights are given by a set of *test functions*. At the idealized population level (corresponding to an infinite sample size), we consider predictors that satisfy the conditions

$$\langle f, \mathcal{B}^\pi Q \rangle_\mu = 0, \qquad \text{for all } f \in \mathcal{F}^\pi. \tag{2}$$

where $\mathcal{F}^\pi$ is a user-defined set of test functions. The two main challenges here are how to use data to enforce an approximate version of such constraints (along with rigorous data-dependent guarantees), and how to design the test function space. We begin with the former challenge.

**Construction of the empirical set:** Given a dataset $\mathcal{D} = \{(s_i, a_i, r_i, s_i^+, o_i)\}_{i=1}^n$, we can approximate the Bellman errors by a linear combination of the temporal difference errors:

$$\int f(s,a) \underbrace{[Q(s,a) - (\mathcal{T}^\pi Q)(s,a)]}_{=\mathcal{B}^\pi Q(s,a)} d\mu \approx \frac{1}{n} \sum_{i=1}^n f(s_i, a_i) \underbrace{[Q(s_i, a_i) - r_i - \gamma Q(s_i^+, \pi)]}_{=\delta^\pi Q(s_i, a_i, r_i, s_i^+, o_i)}. \tag{3}$$

Note that the approximation (3) corresponds to a weighted linear combination of temporal differences. Written more compactly in inner product notation, equation (3) reads $\langle f, \mathcal{B}^\pi Q \rangle_\mu \approx \langle f, \delta^\pi Q \rangle_n$, where $\langle f, g \rangle_n = \frac{1}{n} \sum_{(s,a,r,s^+,o) \in \mathcal{D}} (fg)(s,a,r,s^+,o)$.

In general, the action value function $Q^\pi$ does not satisfy $\langle f, \delta^\pi Q^\pi \rangle_n = 0$ because the empirical approximation (3) involves sampling error. For these reasons, in order to (approximately) identify $Q^\pi$, we impose only inequalities. Given a class of test functions $\mathcal{F}^\pi$, a radius parameter $\rho \geq 0$ and regularization parameter $\lambda \geq 0$, we define the set

$$\widehat{\mathcal{C}}_n^\pi(\rho, \lambda; \mathcal{F}^\pi) \stackrel{def}{=} \left\{ Q \in \mathcal{Q}^\pi \quad \text{such that} \quad \frac{|\langle f, \delta^\pi Q \rangle_n|}{\sqrt{\|f\|_n^2 + \lambda}} \leq \sqrt{\frac{\rho}{n}} \quad \text{for all } f \in \mathcal{F}^\pi \right\}. \tag{4}$$

When the choices of $(\rho, \lambda)$ are clear from the context, we adopt the shorthand $\widehat{\mathcal{C}}_n^\pi(\mathcal{F}^\pi)$, or $\widehat{\mathcal{C}}_n^\pi$ when the function class $\mathcal{F}^\pi$ is also clear. If $\mathcal{F}^\pi$ and $\mathcal{Q}^\pi$ have finite cardinality, $\rho \approx \ln |\mathcal{F}^\pi| |\mathcal{Q}^\pi| + \ln 1/\delta$, where $\delta$ is a prescribed failure probability.

Our definition of the empirical constraint set (4) has two key components: first, the division by $\sqrt{\|f\|_n^2 + \lambda}$ corresponds to a form of *self-normalization*, whereas the addition of $\lambda$ corresponds to a

form of *regularization*. Self-normalization is needed so that the constraints remain suitably scale-invariant. More importantly—in conjunction with the regularization—it ensures that test functions that have poor coverage under the dataset do not have major effects on the solution. In particular, the empirical norm $\|f\|_n^2$ in the self-normalization measures how well the given test function is covered by the dataset. Any test function with poor coverage (i.e., $\|f\|_n^2 \approx 0$) will not yield useful information, and the regularization counteracts its influence. In our guarantees, the choices of $\lambda$ and $\rho$ are critical; as shown in our theory, we typically have $\lambda = \rho/n$, where $\rho$ scales with the metric entropy of the predictor, test and policy spaces. Disregarding $\rho$, the right-hand side of the constraint decays as $1/\sqrt{n}$, so that the constraints are enforced more tightly as the sample size increases.

**Confidence bounds and policy optimization:**  First, for any fixed policy $\pi$, we can use the feasibility set (4) to compute the lower and upper estimates

$$\widehat{V}_{\min}^{\pi} \stackrel{def}{=} \min_{Q \in \widehat{\mathcal{C}}_n^{\pi}(\rho, \lambda; \mathcal{F}^{\pi})} \mathbb{E}_{S \sim \nu_{\text{start}}}\big[Q(S, \pi)\big], \text{ and } \widehat{V}_{\max}^{\pi} \stackrel{def}{=} \max_{Q \in \widehat{\mathcal{C}}_n^{\pi}(\rho, \lambda; \mathcal{F}^{\pi})} \mathbb{E}_{S \sim \nu_{\text{start}}}\big[Q(S, \pi)\big], \quad (5)$$

corresponding to estimates of the minimum and maximum value that the policy $\pi$ can take at the initial distribution. The family of lower estimates can be used to perform policy optimization over the class $\Pi$, in particular by solving the *max-min* problem

$$\max_{\pi \in \Pi} \Big[ \min_{Q \in \widehat{\mathcal{C}}_n^{\pi}} \mathbb{E}_{S \sim \nu_{\text{start}}} Q(S, \pi) \Big], \qquad \text{or equivalently} \qquad \max_{\pi \in \Pi} \widehat{V}_{\min}^{\pi}. \quad (6)$$

**Form of guarantees**  Let us now specify and discuss the types of guarantees that we establish for our estimators (5) and (6). All of our theoretical guarantees involve a $\mu$-based counterpart $\mathcal{C}_n^{\pi}$ of the data-dependent set $\widehat{\mathcal{C}}_n^{\pi}$. More precisely, we define the population set

$$\mathcal{C}_n^{\pi}(4\rho, \lambda; \mathcal{F}^{\pi}) \stackrel{def}{=} \Big\{ Q \in \mathcal{Q}^{\pi} \quad \text{such that} \quad \frac{|\langle f, \mathcal{B}^{\pi} Q \rangle_{\mu}|}{\sqrt{\|f\|_{\mu}^2 + \lambda}} \leq \sqrt{\frac{4\rho}{n}} \qquad \text{for all } f \in \mathcal{F} \Big\}, \quad (7)$$

where $\langle f, g \rangle_{\mu} \stackrel{def}{=} \int f(s,a)g(s,a)d\mu$ is the inner product induced by a distribution[2] $\mu$ over $(s,a)$. As before, we use the shorthand notation $\mathcal{C}_n^{\pi}$ when the underlying arguments are clear from context. Moreover, in the sequel, we generally ignore the constant $4$ in the definition (7) by assuming that $\rho$ is rescaled appropriately—e.g., that we use a factor of $\frac{1}{4}$ in defining the empirical set.

It should be noted that in contrast to the set $\widehat{\mathcal{C}}_n^{\pi}$, the set $\mathcal{C}_n^{\pi}$ is *non-random* and it is defined in terms of the distribution $\mu$ and the input space $(\Pi, \mathcal{F}, \mathcal{Q})$. It relaxes the orthogonality constraints in the weak Bellman formulation (2). Our guarantees for off-policy confidence intervals take the following form:

Coverage guarantee: $\qquad \big[\widehat{V}_{\min}^{\pi}, \widehat{V}_{\max}^{\pi}\big] \ni V^{\pi}.$ $\hspace{3cm}$ (8a)

Width bound: $\qquad \max \Big\{ |\widehat{V}_{\min}^{\pi} - V^{\pi}|, \ |\widehat{V}_{\max}^{\pi} - V^{\pi}| \Big\} \leq \dfrac{1}{1-\gamma} \max_{Q \in \mathcal{C}_n^{\pi}(\mathcal{F}^{\pi})} |\mathbb{E}_{\pi} \mathcal{B}^{\pi} Q|.$ (8b)

Turning to policy optimization, let $\widetilde{\pi}$ be a solution to the max-min criterion (6). Then we prove a result of the following type:

Oracle inequality: $\qquad V^{\widetilde{\pi}} \geq \max_{\pi \in \Pi} \Big\{ \underbrace{V^{\pi}}_{\text{Value}} - \underbrace{\dfrac{1}{1-\gamma} \max_{Q \in \mathcal{C}_n^{\pi}(\mathcal{F})} |\mathbb{E}_{\pi} \mathcal{B}^{\pi} Q|}_{\text{Evaluation uncertainty}} \Big\}.$ (9)

Note that this result guarantees that the estimator competes against an oracle that can search over all policies, and select one based on the optimal trade-off between its value and evaluation uncertainty.

### 3.2 High-probability guarantees

In this section, we present some high-probability guarantees. So as to facilitate understanding under space constraints, we state here results under simplifying assumptions: (a) the dataset originates from a fixed distribution, and (b) the classes $\Pi, \mathcal{F}$ and $\mathcal{Q}$ have finite cardinality. We emphasize that Appendix B provides a far more general version of this result, with an extremely flexible sampling model, and involving metric entropies of parametric or non-parametric function classes.

---

[2]See Section B.2.1 for a precise definition of the relevant $\mu$ for a fairly general sampling model.

**Theorem 1** (Guarantees for finite classes). *Consider a triple $(\Pi, \mathcal{F}, \mathcal{Q})$ that is weakly Bellman realizable (Assumption 2); an i.i.d. dataset (Assumption 1); and the choices $\rho = c\big\{\log(|\mathcal{F}||\Pi||\mathcal{Q}|) + \log(1/\delta)\big\}$ and $\lambda = c'\rho/n$ for some constants $c, c'$. Then w.p. at least $1 - \delta$:*

- *Policy evaluation: For any $\pi \in \Pi$, the estimates $(\widehat{V}^{\pi}_{min}, \widehat{V}^{\pi}_{max})$ specify a confidence interval satisfying the coverage (8a) and width bounds (8b)*

- *Policy optimization: Any max-min policy (6) $\widetilde{\pi}$ satisfies the oracle inequality (9).*

## 4 Concentrability Coefficients and Test Spaces

In this section, we develop some connections to concentrability coefficients that have been used in past work, and discuss various choices of the test class. Like the predictor class $\mathcal{Q}^{\pi}$, the test class $\mathcal{F}^{\pi}$ encodes domain knowledge, and thus its choice is delicate. Different from the predictor class, the test class does not require a 'realizability' condition. As a general principle, the test functions should be chosen as orthogonal as possible with respect to the Bellman residual, so as to enable rapid progress towards the solution; at the same time, they should be sufficiently "aligned" with the dataset, meaning that $\|f\|_{\mu}$ or its empirical counterpart $\|f\|_n$ should be large. Given a test class, each additional test function posits a new constraint which helps identify the correct predictor; at the same time, it increases the metric entropy (parameter $\rho$), which makes each individual constraints more loose. In summary, there are trade-offs to be made in the selection of the test class $\mathcal{F}$, much like $\mathcal{Q}$.

In order to assess the statistical cost that we pay for off-policy data, it is natural to define the *off-policy cost coefficient* (OPC) as

$$K^{\pi}(\mathcal{C}^{\pi}_n, \rho, \lambda) \overset{def}{=} \max_{Q \in \mathcal{C}^{\pi}_n} \frac{|\mathbb{E}_{\pi} \mathcal{B}^{\pi} Q|^2}{(1 + \lambda)\frac{\rho}{n}} = \max_{Q \in \mathcal{C}^{\pi}_n} \frac{\langle \mathbb{1}, \mathcal{B}^{\pi} Q \rangle^2_{\pi}}{(1 + \lambda)\frac{\rho}{n}}, \tag{10}$$

With this notation, our off-policy width bound (8b) can be re-expressed as

$$|\widehat{V}^{\pi}_{\min} - \widehat{V}^{\pi}_{\max}| \leq 2\frac{\sqrt{1+\lambda}}{1-\gamma}\sqrt{K^{\pi}\frac{\rho}{n}}, \tag{11a}$$

while the oracle inequality (9) for policy optimization can be re-expressed in the form

$$V^{\widetilde{\pi}} \geq \max_{\pi \in \Pi}\left\{ V^{\pi} - \frac{\sqrt{1+\lambda}}{1-\gamma}\sqrt{K^{\pi}\frac{\rho}{n}} \right\}, \tag{11b}$$

Since $\lambda \sim \rho/n$, the factor $\sqrt{1 + \lambda}$ can be bounded by a constant in the typical case $n \geq \rho$. We now offer concrete examples of the OPC , while deferring further examples to Appendix A.5.

### 4.1 Likelihood ratios

Our broader goal is to obtain small Bellman error along the distribution induced by $\pi$. Assume that one constructs a test function class $\mathcal{F}^{\pi}$ of possible likelihood ratios.

**Proposition 1** (Likelihood ratio bounds). *Assume that for some constant $b_{\pi}$, the test function defined as $f^*(s, a) = \frac{1}{b_{\pi}}\frac{d_{\pi}(s,a)}{\mu(s,a)}$ belongs to $\mathcal{F}^{\pi}$ and satisfies $\|f^*\|_{\infty} \leq 1$. Then the OPC coefficient satisfies*

$$K^{\pi} \overset{(i)}{\leq} \frac{\mathbb{E}_{\pi}\left[\frac{d_{\pi}(S,A)}{\mu(S,A)}\right] + b_{\pi}^2\lambda}{1+\lambda} \overset{(ii)}{\leq} \frac{b_{\pi}\left(1 + b_{\pi}\lambda\right)}{1+\lambda}. \tag{12}$$

Here $b_{\pi}$ is a scaling parameter that ensures $\|f^*\|_{\infty} \leq 1$. Concretely one can take $b_{\pi} = \sup_{(s,a)} \frac{d_{\pi}(s,a)}{\mu(s,a)}$. The proof is in Appendix D.1. Since $\lambda = \lambda_n \to 0$ as $n$ increases, the OPC coefficient is bounded by a multiple of the expected ratio $\mathbb{E}_{\pi}\left[\frac{d_{\pi}(S,A)}{\mu(S,A)}\right]$. Up to an additive offset, this expectation is equivalent to the $\chi^2$-distribution between the policy-induced occupation measure $d_{\pi}$ and data-generating distribution $\mu$. The concentrability coefficient can be plugged back into Eqs. (11a) and (11b) to obtain a concrete policy optimization bound. In this case, we recover a result similar to [XJ20b], but with a much milder concentrability coefficient that involves only the chosen comparator policy.

## 4.2 The error test space

We now turn to the discussion of a choice for the test space that extends the LSTD algorithm to non-linear spaces. A simplification to the linear setting is presented later in Section 5.

As is well known, the LSTD algorithm [BB96] can be seen as minimizing the Bellman error projected onto the linear prediction space $Q$. Define the transition operator $(\mathbb{P}^\pi Q)(s,a) = \mathbb{E}_{s^+ \sim \mathbb{P}(s,a)} Q(s^+, \pi)$, and the prediction error $\epsilon = Q - Q_\star^\pi$, where $Q_\star^\pi$ is a $Q$-function from the definition of weak realizability. The Bellman error can be re-written as $\mathcal{B}^\pi Q = \mathcal{B}^\pi Q - \mathcal{B}^\pi Q_\star^\pi = (\mathcal{I} - \gamma \mathbb{P}^\pi)\epsilon$. When realizability holds, in the linear setting and at the population level, the LSTD solution seeks to satisfy the projected Bellman equations

$$\langle f, \mathcal{B}^\pi Q \rangle_\mu = 0, \quad \text{for all } f \in \mathcal{E}_\star^\pi. \tag{13}$$

In the linear case, $\mathcal{E}_\star^\pi$ is the class of linear functions $\mathcal{Q}^\pi$ used as predictors; when $\mathcal{Q}^\pi$ is non-linear, we can extend the LSTD method by using the (nonlinear) error test space $\mathcal{F}^\pi = \mathcal{E}_\star^\pi = \{Q - Q_\star^\pi\}$. Since $\mathcal{E}_\star^\pi$ is unknown (as it depends on the weak solution $Q_\star^\pi$), we choose instead the larger class

$$\mathcal{E}^\pi = \{Q - Q' \mid Q, Q' \in \mathcal{Q}^\pi\},$$

which contains $\mathcal{E}_\star^\pi$. The resulting approach can be seen as performing a projection of the Bellman operator $\mathcal{B}^\pi Q$ into the error space $\mathcal{E}_\star^\pi$, much like LSTD does in the linear setting. However, different from LSTD, our procedure returns confidence intervals as opposed to a point estimator. This choice of the test space is related to the Bubnov-Galerkin method [Rep17] for linear spaces; it selects the test space $\mathcal{F}^\pi$ to be identical to the trial space $\mathcal{E}_\star^\pi$ that contains all possible solution errors.

**Lemma 1** (OPC coefficient from prediction error). *For any test function class $\mathcal{F}^\pi \supseteq \mathcal{E}^\pi$, we have*

$$K^\pi \le \max_{Q \in \mathcal{Q}^\pi} \left\{ \frac{\|\epsilon\|_\mu^2 + \lambda}{\|\mathbb{1}\|_\pi^2 + \lambda} \frac{\langle \mathbb{1}, \mathcal{B}^\pi Q \rangle_\pi^2}{\langle \epsilon, \mathcal{B}^\pi Q \rangle_\mu^2} \right\} = \max_{\epsilon \in \mathcal{E}_\star^\pi} \left\{ \frac{\|\epsilon\|_\mu^2 + \lambda}{\|\mathbb{1}\|_\pi^2 + \lambda} \frac{\langle \mathbb{1}, (\mathcal{I} - \gamma \mathbb{P}^\pi)\epsilon \rangle_\pi^2}{\langle \epsilon, (\mathcal{I} - \gamma \mathbb{P}^\pi)\epsilon \rangle_\mu^2} \right\}. \tag{14}$$

The above coefficient measures the ratio between the Bellman error along the distribution of the target policy $\pi$ and that projected onto the error space $\mathcal{E}_\star^\pi$ defined by $\mathcal{Q}^\pi$. It is a concentrability coefficient that *always* applies, as the choice of the test space does not require domain knowledge. See Appendix D.2 for the proof, and Appendix A.6 for further comments and insights, as well as a simplification in the special case of Bellman closure.

## 4.3 The Bellman test space

In the prior section we controlled the projected Bellman error. Another longstanding approach in reinforcement learning is to control the Bellman error itself, for example by minimizing the squared Bellman residual. In general, this cannot be done if only an offline dataset is available due to the well known *double sampling* issue. However, in some cases we can use an helper class to try to capture the Bellman error. Such class needs to be a superset of the class of *Bellman test functions* given by

$$\mathcal{F}_\pi^{\mathcal{B}} \overset{def}{=} \{\mathcal{B}^\pi Q \mid Q \in \mathcal{Q}^\pi\}. \tag{15}$$

Any test class that contains the above allows us to control the Bellman residual, as we show next.

**Lemma 2** (Bellman Test Functions). *For any test function class $\mathcal{F}^\pi$ that contains $\mathcal{F}_\pi^{\mathcal{B}}$, we have*

$$\|\mathcal{B}^\pi Q\|_\mu \le c_1 \sqrt{\frac{\rho}{n}} \quad \text{for any } Q \in \mathcal{C}_n^\pi(\mathcal{F}^\pi). \tag{16a}$$

*Moreover, the off-policy cost coefficient is upper bounded as*

$$K^\pi \overset{(i)}{\le} c_1 \sup_{Q \in \mathcal{Q}^\pi} \frac{\langle \mathbb{1}, \mathcal{B}^\pi Q \rangle_\pi^2}{\|\mathcal{B}^\pi Q\|_\mu^2} \overset{(ii)}{\le} c_1 \sup_{Q \in \mathcal{Q}^\pi} \frac{\|\mathcal{B}^\pi Q\|_\pi^2}{\|\mathcal{B}^\pi Q\|_\mu^2} \overset{(iii)}{\le} c_1 \sup_{(s,a)} \frac{d_\pi(s,a)}{\mu(s,a)}. \tag{16b}$$

See Appendix D.4 for the proof of this claim.

Consequently, whenever the test class includes the Bellman test functions, the off-policy cost coefficient is at most the ratio between the squared Bellman residuals along the data generating distribution and the target distribution. If Bellman closure holds, then the prediction error space $\mathcal{E}^\pi$ introduced in Section 4.2 contains the Bellman test functions: for $Q \in \mathcal{Q}^\pi$, we can write $\mathcal{B}^\pi Q = Q - \mathcal{T}^\pi Q \in \mathcal{E}^\pi$. This fact allows us to recover a result in the recent paper [XCJ+21] in the special case of Bellman closure, although the approach presented here is more general.

### 4.4 Combining test spaces

Often, it is natural to construct a test space that is a union of several simpler classes. A simple but valuable observation is that the resulting procedure inherits the best of the OPC coefficients. Suppose that we are given a collection $\{\mathcal{F}_m^\pi\}_{m=1}^M$ of $M$ different test function classes, and define the union $\mathcal{F}^\pi = \bigcup_{m=1}^M \mathcal{F}_m^\pi$. For each $m = 1, \ldots, M$, let $K_m^\pi$ be the OPC coefficient defined by the function class $\mathcal{F}_m^\pi$ and radius $\rho$, and let $K^\pi(\mathcal{F})$ be the OPC coefficient associated with the full class. Then we have the following guarantee:

**Lemma 3** (Multiple test classes). $K^\pi(\mathcal{F}) \leq \min_{m=1,\ldots,M} K_m^\pi$.

This guarantee is a straightforward consequence of our construction of the feasibility sets: in particular, we have $\mathcal{C}_n^\pi(\mathcal{F}) = \cap_{m=1}^M \mathcal{C}_n^\pi(\mathcal{F}_m)$, and consequently, by the variational definition of the off-policy cost coefficient $K^\pi(\mathcal{F})$ as optimization over $\mathcal{C}_n^\pi(\mathcal{F})$, the bound (3) follows. In words, when multiple test spaces are combined, then our algorithms inherit the best (smallest) OPC coefficient over all individual test spaces. While this behavior is attractive, one must note that there is a statistical cost to using a union of test spaces: the choice of $\rho$ scales as a function of $\mathcal{F}$ via its metric entropy. This increase in $\rho$ must be balanced with the benefits of using multiple test spaces.[3]

## 5 Linear Setting

In this section, we turn to a detailed analysis of our estimators using function classes that are linear in a feature map. Let $\phi : \mathcal{S} \times \mathcal{A} \to \mathbb{R}^d$ be a given feature map, and consider linear expansions $g_w(s, a) \stackrel{def}{=} \langle w, \phi(s, a) \rangle = \sum_{j=1}^d w_j \phi_j(s, a)$. The class of *linear functions* takes the form

$$\mathcal{L} \stackrel{def}{=} \{(s, a) \mapsto g_w(s, a) \mid w \in \mathbb{R}^d, \|w\|_2 \leq 1\}. \tag{17}$$

Throughout our analysis, we assume that $\|\phi(s, a)\|_2 \leq 1$ for all state-action pairs.

Following the approach in Section 4.2, which is based on the LSTD method, we should choose the test function class $\mathcal{F}^\pi = \mathcal{L}$, as in the linear case the prediction error is linear.

In order to obtain a computationally efficient implementation, we need to use a test class that is a "simpler" subset of $\mathcal{L}$. In particular, for linear functions, it is not hard to show that the estimates $\widehat{V}_{\min}^\pi$ and $\widehat{V}_{\max}^\pi$ from equation (5) can be computed by solving a quadratic program, with two linear constraints for each test function. (See Appendix A.8 for the details.) Consequently, the computational complexity scales linearly with the number of test functions. Thus, if we restrict ourselves to a finite test class contained within $\mathcal{L}$, we will obtain a computationally efficient approach.

### 5.1 A computationally friendly test class and OPC coefficients

Define the empirical covariance matrix $\widehat{\Sigma} = \frac{1}{n} \sum_{i=1}^n \phi_i \phi_i^T$ where $\phi_i \stackrel{def}{=} \phi(s_i, a_i)$. Let $\{\widehat{u}_j\}_{j=1}^d$ be the eigenvectors of empirical covariance matrix $\widehat{\Sigma}$, and suppose that they are normalized to have unit $\ell_2$-norm. We use these normalized eigenvectors to define the finite test class

$$\widetilde{\mathcal{F}}^\pi \stackrel{def}{=} \{f_j, j = 1, \ldots, d\} \quad \text{where } f_j(s, a) \stackrel{def}{=} \langle \widehat{u}_j, \phi(s, a) \rangle \tag{18}$$

A few observations are in order:

- This test class has only $d$ functions, so that our QP implementation has $2d$ constraints, and can be solved in polynomial time. (Again, see Appendix A.8 for details.)

- Since $\widetilde{\mathcal{F}}^\pi$ is a subset of $\mathcal{L}$ the choice of radius $\rho = c(\frac{d}{n} + \log 1/\delta)$ is valid for some constant $c$.

**Concentrability:** When weak Bellman closure does not hold, then our analysis needs to take into account how errors propagate via the dynamics. In particular, we define the *next-state feature extractor* $\phi^{+\pi}(s, a) \stackrel{def}{=} \mathbb{E}_{s^+ \sim \mathbb{P}(s,a)} \phi(s^+, \pi)$, along with the population covariance matrix $\Sigma \stackrel{def}{=} \mathbb{E}_\mu[\phi(s, a)\phi^\top(s, a)]$, and its $\lambda$-regularized version $\Sigma_\lambda \stackrel{def}{=} \Sigma + \lambda I$. We also define the matrices

$$\Sigma^{+\pi} \stackrel{def}{=} \mathbb{E}_\mu[\phi(\phi^{+\pi})^\top], \quad \Sigma_{\lambda,\text{Boot}}^{+\pi} \stackrel{def}{=} (\Sigma_\lambda^{\frac{1}{2}} - \gamma \Sigma_\lambda^{-\frac{1}{2}} \Sigma^{+\pi})^\top (\Sigma_\lambda^{\frac{1}{2}} - \gamma \Sigma_\lambda^{-\frac{1}{2}} \Sigma^{+\pi}).$$

---

[3]For space reasons, we defer to Appendix A.7 an application in which we construct a test function space as a union of subclasses, and thereby obtain a method that automatically leverages Bellman closure when it holds, falls back to importance sampling if closure fails, and falls back to a worst-case bound in general.

The matrix $\Sigma^{+\pi}$ is the cross-covariance between successive states, whereas the matrix $\Sigma^{+\pi}_{\lambda,\text{Boot}}$ is a suitably renormalized and symmetrized version of the matrix $\Sigma^{\frac{1}{2}} - \gamma \Sigma^{-\frac{1}{2}} \Sigma^{+\pi}$, which arises naturally from the policy evaluation equation. We refer to quantities that contain evaluations at the next-state (e.g., $\phi^{+\pi}$) as bootstrapping terms, and now bound the OPC coefficient in the presence of such terms:

**Proposition 2** (OPC bounds with bootstrapping). *Under weak realizability, we have*

$$K^\pi(\widetilde{\mathcal{F}}^\pi) \le c\, d \|\mathbb{E}_\pi[\phi - \gamma\phi^{+\pi}]\|^2_{(\Sigma^{+\pi}_{\lambda,Boot})^{-1}} \qquad \text{with probability at least } 1 - \delta. \tag{19}$$

See Appendix E.1 for the proof. The bound (19) takes a familiar form, as it involves the same matrices used to define the LSTD solution. This is expected, as our approach here is essentially equivalent to the LSTD method; the difference is that LSTD only gives a point estimate as opposed to the confidence intervals that we present here; however, they are both derived from the same principle, namely from the Bellman equations projected along the predictor (error) space.

The bound quantifies how the feature extractor $\phi$ together with the bootstrapping term $\phi^{+\pi}$, averaged along the target policy $\pi$, interact with the covariance matrix with bootstrapping $\Sigma^{+\pi}_{\lambda,\text{Boot}}$. It is an approximation to the OPC coefficient bound derived in Lemma 1. The bootstrapping terms capture the temporal difference correlations that can arise in reinforcement learning when strong assumptions like Bellman closure do not hold. As a consequence, such an OPC coefficient being small is a *sufficient* condition for reliable off-policy prediction. This bound on the OPC coefficient always applies, and it reduces to the simpler one (20) when weak Bellman closure holds, with no need to inform the algorithm of the simplified setting; see Appendix E.3 for the proof.

**Proposition 3** (OPC bounds under weak Bellman Closure). *Under Bellman closure, we have*

$$K^\pi(\widetilde{\mathcal{F}}^\pi) \le c\, d \|\mathbb{E}_\pi\phi\|^2_{\Sigma_\lambda^{-1}} \qquad \text{with probability at least } 1 - \delta. \tag{20}$$

## 5.2 Actor-critic scheme for policy optimization

Having described a practical procedure to compute $\widehat{V}^\pi_{\min}$, we now turn to the computation of the max-min estimator for policy optimization. We define the *soft-max policy class*

$$\Pi_{\text{lin}} \overset{def}{=} \left\{ (s, a) \mapsto \frac{e^{\langle\phi(s,a),\theta\rangle}}{\sum_{a^+ \in \mathcal{A}} e^{\langle\phi(s,a^+),\theta\rangle}} \mid \|\theta\|_2 \le T,\ \theta \in \mathbb{R}^d \right\}. \tag{21}$$

In order to compute the max-min solution (6) over this policy class, we implement an actor-critic method, in which the actor performs a variant of mirror descent.[4]

- At each iteration $t = 1, \ldots, T$, the policy $\pi_t \in \Pi_{\text{lin}}$ can be identified with a parameter $\theta_t \in \mathbb{R}^d$. The sequence is initialized with $\theta_1 = 0$.

- Using the finite test function class (18) based on normalized eigenvectors, the pessimistic value estimate $\widehat{V}^{\pi_t}_{\min}$ is computed by solving a quadratic program, as previously described. This computation returns the weight vector $w_t$ of the associated optimal action-value function.

- Using the action-value vector $w_t$, we update the actor's parameter as

$$\theta_{t+1} = \theta_t + \eta w_t \qquad \text{where } \eta = \sqrt{\frac{\log|\mathcal{A}|}{2T}} \text{ is a stepsize parameter.} \tag{22}$$

We now state a guarantee on the behavior of this procedure, based on two OPC coefficients:

$$K^{\widetilde{\pi}}_{(1)} = d \|\mathbb{E}_{\widetilde{\pi}}\phi\|^2_{\Sigma_\lambda^{-1}}, \quad \text{and} \quad K^{\widetilde{\pi}}_{(2)} = d \sup_{\pi \in \Pi} \left\{ \|\mathbb{E}_{\widetilde{\pi}}[\phi - \gamma\phi^{+\pi}]\|^2_{(\Sigma^{+\pi}_{\lambda,Boot})^{-1}} \right\}. \tag{23}$$

Moreover, in making the following assertion, we assume that every weak solution $Q^\pi_\star$ can be evaluated against the distribution of a comparator policy $\widetilde{\pi} \in \Pi$, i.e., $\langle \mathbb{1}, \mathcal{B}^\pi Q^\pi_\star \rangle_{\widetilde{\pi}} = 0$ for all $\pi \in \Pi$. (This assumption is still weaker than strong realizability).

**Theorem 2** (Approximate Guarantees for Linear Soft-Max Optimization). *Under the above conditions, running the procedure for $T$ rounds returns a policy sequence $\{\pi_t\}_{t=1}^T$ such that, for any comparator policy $\widetilde{\pi} \in \Pi$,*

$$\frac{1}{T}\sum_{t=1}^T \left\{ V^{\widetilde{\pi}} - V^{\pi_t} \right\} \le \frac{c_1}{1-\gamma} \left\{ \underbrace{\sqrt{\frac{\log|\mathcal{A}|}{T}}}_{\text{Optimization error}} + \underbrace{\sqrt{K^{\widetilde{\pi}}_{(\cdot)} \frac{d\log(nT) + \log\left(\frac{n}{\delta}\right)}{n}}}_{\text{Statistical error}} \right\}, \tag{24}$$

---
[4]Strictly speaking, it is mirror ascent, but we use the conventional terminology.

*with probability at least* $1 - \delta$. *This bound always holds with* $K^{\widetilde{\pi}}_{(\cdot)} = K^{\widetilde{\pi}}_{(2)}$, *and moreover, it holds with* $K^{\widetilde{\pi}}_{(\cdot)} = K^{\widetilde{\pi}}_{(1)}$ *when weak Bellman closure is in force.*

See Appendix F for the proof. Whenever Bellman closure holds, the result automatically inherits the more favorable concentrability coefficient $K^{\widetilde{\pi}}_{(2)}$, as originally derived in Proposition 3. The resulting bound is only $\sqrt{d}$ worse than the lower bound recently established in the paper [ZWB21]. However, the method proposed here is robust, in that it provides guarantees even when Bellman closure does not hold. In this case, we have a guarantee in terms of the OPC coefficient $K^{\widetilde{\pi}}_{(1)}$. Note that it is a uniform version of the one derived previously in Proposition 2, in that there is an additional supremum over the policy class. This supremum arises due to the use of gradient-based method, which implicitly searches over policies in bootstrapping terms; see Appendix A.9 for a more detailed discussion of this issue.

## Acknowledgment

AZ was partially supported by NSF-FODSI grant 2023505. In addition, this work was partially supported by NSF-DMS grant 2015454, NSF-IIS grant 1909365, as well as Office of Naval Research grant DOD-ONR-N00014-18-1-2640 to MJW. The authors are grateful to Nan Jiang and Alekh Agarwal for pointing out further connections with the existing literature, as well as to the reviewers for pointing out clarity issues.

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
