# Bellman Residual Orthogonalization
# for Offline Reinforcement Learning

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

# Contents

# A  Additional Discussion and Results

## A.1  Bellman Residual Orthogonalization

Suppose that our goal is to estimate the action-value function $Q^\pi$ of a given policy $\pi$. This function is known to be a fixed point of the Bellman evaluation operator $\mathcal{T}^\pi$ associated with the policy $\pi$. Thus, when the MDP is known, one option is to (approximately) solve the Bellman evaluation equations $Q(s,a) = (\mathcal{T}^\pi Q)(s,a)$ for all state-action pairs. However, even if function approximation for $Q$ is implemented, it is still difficult to directly solve these equations if the state-action space is sufficiently complex.

This observation motivates the strategy taken in this paper: instead of enforcing the Bellman equations for all state-action pairs, suppose that we do so only in an average sense, and with respect to a certain set of functions. More formally, a *test function* is a mapping from the state-action space to the real line; any such function serves to enforce the Bellman equations in an average sense in the following way. Let $\mathcal{F}^\pi$ denote some user-prescribed class of test functions, which we refer to as the *test space*. Then for a given measure $\mu$, we require only that the action-value function $Q^\pi$ satisfy the integral constraints

$$\langle f, Q - \mathcal{T}^\pi(Q)\rangle_\mu \stackrel{def}{=} \int f(s,a)[Q(s,a) - (\mathcal{T}^\pi Q)(s,a)]d\mu = 0, \qquad \text{for all } f \in \mathcal{F}^\pi. \tag{25}$$

We refer to this design principle as *Bellman residual orthogonalization*, because it requires the Bellman error function to be orthogonal to a set of test functions, as measured under the $L^2(\mu)$ inner product. Of course, by enlarging the test space $\mathcal{F}^\pi$, the Bellman error is required to be orthogonal to more test functions, and it will ultimately be zero if enough test functions are added as constraints. But at the same time, as shown by our analysis, any such enlargement has both computational and statistical costs, so there are tradeoffs to be understood.

In numerical analysis, especially in solving partial differential equations, the design principle (25) is called the weak or variational formulation (e.g., [Eva10]), and its solutions are referred to as weak solutions. Here we are advocating a *weak formulation* of the Bellman equations. Of course, the constraints (25) are necessary but not sufficient: the *weak (Bellman) solutions* need not solve the Bellman equations. However, whenever we need to learn based on a limited dataset, it is unreasonable to satisfy the Bellman equations everywhere; instead, by choosing the test space appropriately, we can seek to satisfy the Bellman equations over regions of the state-action space that are most important. In some cases, the formulation (25) can be fruitfully viewed as a type of Galerkin approximation (e.g., [Gal15, Fle84]) to the Bellman equations. For example, when both the test functions and $Q$-value functions belong to some linear space (and the empirical constraints are enforced exactly), then the weak formulation and Galerkin approximation lead to the least-squares temporal difference (LSTD) estimator; this connection between Galerkin methods and LSTD has been noted in past work by Yu and Bertsekas [YB10]. In this paper, our goal is to understand the weak formulation (25) in a broader sense for general test and predictor classes.

## A.2  Comparison with Weight Learning Methods

The work closest to ours is [JH20]. They also use an auxiliary weight function class, which is comparable to our test class. However, the test class is used in different ways; we compare them in this section at the population level.[5] Let us assume that weak realizability holds and that $\mathcal{F}$ is symmetric, i.e., if $f \in \mathcal{F}$ then $-f \in \mathcal{F}$ as well. At the population level, our program seeks to solve

$$\sup_{Q \in \mathcal{Q}^\pi} \mathbb{E}_{s\sim\nu_{\text{start}}} Q(s,\pi) \quad \text{s.t.} \quad \sup_{f\in\mathcal{F}} \langle f, \mathcal{B}^\pi Q\rangle_\mu = 0, \tag{26}$$

which is equivalent for any $w \in \mathcal{F}$ to

$$\sup_{Q \in \mathcal{Q}^\pi} \mathbb{E}_{s\sim\nu_{\text{start}}} Q(s,\pi) - \frac{1}{1-\gamma}\langle w, \mathcal{B}^\pi Q\rangle_\mu \quad \text{s.t.} \quad \sup_{f\in\mathcal{F}} \langle f, \mathcal{B}^\pi Q\rangle_\mu = 0.$$

---

[5]The empirical estimator in [JH20] does not take into account the 'alignment' of each weight function with respect to the dataset, which we do through self-normalization and regularization in the construction of the empirical estimator. This precludes obtaining the same type of strong finite time guarantees that we are able to derive here.

Removing the constraints leads to the upper bound

$$\sup_{Q \in \mathcal{Q}^\pi} \mathbb{E}_{s \sim \nu_{\text{start}}} Q(s, \pi) - \frac{1}{1-\gamma} \langle w, \mathcal{B}^\pi Q \rangle_\mu.$$

Since this is a valid upper bound for any $w \in \mathcal{F}$, minimizing over $w$ must still yield an upper bound, which reads

$$\inf_{w \in \mathcal{F}} \sup_{Q \in \mathcal{Q}^\pi} \mathbb{E}_{s \sim \nu_{\text{start}}} Q(s, \pi) - \frac{1}{1-\gamma} \langle w, \mathcal{B}^\pi Q \rangle_\mu.$$

This is the population program for "weight learning", as described in [JH20]. It follows that Bellman residual orthogonalization always produces tighter confidence intervals than "weight learning" at the population level.

Another interesting comparison is with "value learning", also described in [JH20]. In this case, assuming symmetric $\mathcal{F}$, we can equivalently express the population program (26) using a Lagrange multiplier as follows

$$\sup_{Q \in \mathcal{Q}^\pi} \mathbb{E}_{s \sim \nu_{\text{start}}} Q(s, \pi) - \sup_{\lambda \geq 0, f \in \mathcal{F}} \lambda \langle f, \mathcal{B}^\pi Q \rangle_\mu. \tag{27}$$

Rearranging we obtain

$$\sup_{Q \in \mathcal{Q}^\pi} \inf_{\lambda \geq 0, f \in \mathcal{F}} \mathbb{E}_{s \sim \nu_{\text{start}}} Q(s, \pi) - \lambda \langle f, \mathcal{B}^\pi Q \rangle_\mu.$$

The "value learning" program proposed in [JH20] has a similar formulation to ours but differs in two key aspects. The first—and most important—is that [JH20] ignores the Lagrange multiplier; this means "value learning" is not longer associated to a constrained program. While the Lagrange multiplier could be "incorporated" into the test class $\mathcal{F}$, doing so would cause the entropy of $\mathcal{F}$ to be unbounded. Another point of difference is that "value learning" uses such expression with $\lambda = 1$ to derive the confidence interval *lower bound*, while we use it to construct the confidence interval *upper bound*. While this may seem like a contradiction, we notice that the expression is derived using different assumptions: we assume weak realizability of $Q$, while [JH20] assumes realizability of the density ratios between $\mu$ and the discounted occupancy measure $\pi$.

## A.3 Additional Literature

Here we summarize some additional literature. The efficiency of off-policy tabular RL has been investigated in the papers [YBW20, YW20, YW21]. For empirical studies on offline RL, see the papers [LTDC19, JGS+19, WTN19, ASN20, WNŻ+20, SSB+20, NDGL20, YQCC21, KHSL21, BGB20, KFTL19, KRNJ20, YTY+20].

Some of the classical RL algorithm are presented in the papers [Mun03, Mun05, AMS07, ASM08, FSM10, FGSM16]. For a more modern analysis, see [CJ19]. These works generally make additionally assumptions on top of realizability. Alternatively, one can use importance sampling [Pre00, TB16, JL16, FCG18]. A more recent idea is to look at the distributions themselves [LLTZ18, NDK+19, XMW19, ZDLS20, ZLW20, YND+20, KU19].

Offline policy optimization with pessimism has been studied in the papers [LSAB20, RZM+21, JYW21, XCJ+21, ZWB21, YWDW, US21]. There exists a fairly extensive literature on lower bounds with linear representations, including the two papers [Zan20, WFK20] that concurrently derived the first exponential lower bounds for the offline setting, and [FKSLX21] proves that realizability and coverage alone are insufficient.

In the context of off-policy optimization several works have investigated methods that assume only realizability of the optimal policy [XJ20a, XJ20b]. Related work includes the papers [DW20, DJL21, JH20, UHJ20, TFL+19, ND20, VJY21, HJD+21, ZSU+22, UIJ+21, CQ22, LTND21]. Among concurrent works, we note [ZHH+22].

## A.4 Definition of Weak Bellman Closure

**Definition 1** (Weak Bellman Closure). *The Bellman operator $\mathcal{T}^\pi$ is weakly closed with respect to the triple $(\mathcal{Q}^\pi, \mathcal{F}^\pi, \mu)$ if for any $Q \in \mathcal{Q}^\pi$, there exists a predictor $\mathcal{P}^\pi(Q) \in \mathcal{Q}^\pi$ such that*

$$\langle f, \mathcal{P}^\pi(Q) \rangle_\mu = \langle f, \mathcal{T}^\pi(Q) \rangle_\mu. \tag{28}$$

### A.5 Additional results on the concentrability coefficients

### A.5.1 Testing with the identity function

Suppose that the identity function $\mathbb{1}$ belongs to the test class. Doing so amounts to requiring that the Bellman error is controlled in an average sense over all the data. When this choice is made, we can derive some generic upper bounds on $K^\pi$, which we state and prove here:

**Lemma 4.** *If $\mathbb{1} \in \mathcal{F}^\pi$, then we have the upper bounds*

$$K^\pi \overset{(i)}{\leq} \frac{\max_{Q \in \mathcal{C}_n^\pi} |\mathbb{E}_\pi \mathcal{B}^\pi Q|^2}{\max_{Q \in \mathcal{C}_n^\pi} |\mathbb{E}_\mu \mathcal{B}^\pi Q|^2} \overset{(ii)}{\leq} K_*^\pi \overset{def}{=} \max_{Q \in \mathcal{C}_n^\pi} \frac{|\mathbb{E}_\pi \mathcal{B}^\pi Q|^2}{|\mathbb{E}_\mu \mathcal{B}^\pi Q|^2}. \tag{29}$$

*Proof.* Since $\mathbb{1} \in \mathcal{F}$, the definition of $\mathcal{C}_n^\pi$ implies that

$$\max_{Q \in \mathcal{C}_n^\pi} |\mathbb{E}_\mu \mathcal{B}^\pi Q|^2 \leq \left( \| \mathbb{1} \|_\mu^2 + \lambda \right) \frac{\rho}{n} = \left( 1 + \lambda \right) \frac{\rho}{n}.$$

The upper bound (i) then follows from the definition of $K^\pi$. The upper bound (ii) follows since the right hand side is the maximum ratio. $\qquad \square$

Note that large values of $K_*^\pi$ can arise when there exist $Q$-functions in the set $\mathcal{C}_n^\pi$ that have low average Bellman error under the data-generating distribution $\mu$, but relatively large values under $\pi$. Of course, the likelihood of such unfavorable choices of $Q$ is reduced when we use a larger test function class, which then reduces the size of $\mathcal{C}_n^\pi$. However, we pay a price in choosing a larger test function class, since the choice (40b) of the radius $\rho$ needed for Theorem 3 depends on its complexity.

### A.5.2 Mixture distributions

Now suppose that the dataset consists of a collection of trajectories collected by different protocols. More precisely, for each $j = 1, \ldots, m$, let $\mu_j$ be a particular protocol for generating a trajectory. Suppose that we generate data by first sampling a random index $J \in [m]$ according to a probability distribution $\{p_j\}_{j=1}^m$, and conditioned $J = j$, we sample $(s, a, o)$ according to $\mu_j$. The resulting data follows a mixture distribution, where we set $o = j$ to tag the protocol used to generate the data. To be clear, for each sample $i = 1, \ldots, n$, we sample $J$ as described, and then draw a single sample $(s, a, o) \sim \mu_j$.

Following the intuition given in the previous section, it is natural to include test functions that code for the protocol—that is, the binary-indicator functions

$$f_j(s, a, o) = \begin{cases} 1 & \text{if } o = j \\ 0 & \text{otherwise.} \end{cases} \tag{30}$$

This test function, when included in the weak formulation, enforces the Bellman evaluation equations for the policy $\pi \in \Pi$ under consideration along the distribution induced by each data-generating policy $\mu_j$.

**Lemma 5** (Mixture Policy Concentrability). *Suppose that $\mu$ is an $m$-component mixture, and that the indicator functions $\{f_j\}_{j=1}^m$ are included in the test class. Then we have the upper bounds*

$$K^\pi \overset{(i)}{\leq} \frac{1 + m\lambda}{1 + \lambda} \frac{\max_{Q \in \mathcal{C}_n^\pi} [\mathbb{E}_\pi \mathcal{B}^\pi Q]^2}{\max_{Q \in \mathcal{C}_n^\pi} \sum_{j=1}^m p_j^2 [\mathbb{E}_{\mu_j} \mathcal{B}^\pi Q]^2} \overset{(ii)}{\leq} \frac{1 + m\lambda}{1 + \lambda} \max_{Q \in \mathcal{C}_n^\pi} \left\{ \frac{[\mathbb{E}_\pi \mathcal{B}^\pi Q]^2}{\sum_{j=1}^m p_j^2 [\mathbb{E}_{\mu_j} \mathcal{B}^\pi Q]^2} \right\}. \tag{31}$$

*Proof.* From the definition of $K^\pi$, it suffices to show that

$$\max_{Q \in \mathcal{C}_n^\pi} \sum_{j=1}^m p_j^2 [\mathbb{E}_{\mu_j} \mathcal{B}^\pi Q]^2 \leq \frac{\rho}{n} \left( 1 + m\lambda \right).$$

A direct calculation yields $\langle f_j, \mathcal{B}^\pi Q \rangle_\mu = \mathbb{E}_\mu \mathbb{I}\{o = j\} \mathcal{B}^\pi Q = p_j \mathbb{E}_{\mu_j} \mathcal{B}^\pi Q$. Moreover, since each $f_j$ belongs to the test class by assumption, we have the upper bound $\left| p_j \mathbb{E}_{\mu_j} \mathcal{B}^\pi Q \right| \leq \sqrt{\frac{\rho}{n}} \sqrt{\|f_j\|_\mu^2 + \lambda}$.

Squaring each term and summing over the constraints yields

$$\sum_{j=1}^{m} p_j^2 [\mathbb{E}_{\mu_j} \mathcal{B}^\pi Q]^2 \leq \frac{\rho}{n} \sum_{j=1}^{m} \left( \|f_j\|_\mu^2 + \lambda \right) = \frac{\rho}{n} \left( 1 + m\lambda \right),$$

where the final equality follows since $\sum_{j=1}^{m} \|f_j\|_\mu^2 = 1$. $\qquad\square$

As shown by the upper bound, the off-policy coefficient $K^\pi$ provides a measure of how the squared-averaged Bellman errors along the policies $\{\mu_j\}_{j=1}^m$, weighted by their probabilities $\{p_j\}_{j=1}^m$, transfers to the evaluation policy $\pi$. Note that the regularization parameter $\lambda$ decays as a function of the sample size—e.g., as $1/n$ in Theorem 3—the factor $(1+m\lambda)/(1+\lambda)$ approaches one as $n$ increases (for a fixed number $m$ of mixture components).

### A.5.3 Bellman Rank for off-policy evaluation

In this section, we show how more refined bounds can be obtained when—in addition to a mixture condition—additional structure is imposed on the problem. In particular, we consider a notion similar to that of Bellman rank [JKA$^+$17], but suitably adapted[6] to the off-policy setting.

Given a policy class $\widetilde{\Pi}$ and a predictor class $\widetilde{\mathcal{Q}}$, we say that it has Bellman rank is $d$ if there exist two maps $\nu : \widetilde{\Pi} \to \mathbb{R}^d$ and $\xi : \widetilde{\mathcal{Q}} \to \mathbb{R}^d$ such that

$$\mathbb{E}_\pi \mathcal{B}^\pi Q = \langle \nu_\pi, \xi_Q \rangle_{\mathbb{R}^d}, \qquad \text{for all } \pi \in \widetilde{\Pi} \text{ and } Q \in \widetilde{\mathcal{Q}}. \tag{32}$$

In words, the average Bellman error of any predictor $Q$ along any given policy $\pi$ can be expressed as the Euclidean inner product between two $d$-dimensional vectors, one for the policy and one for the predictor. As in the previous section, we assume that the data is generated by a mixture of $m$ different distributions (or equivalently policies) $\{\mu_j\}_{j=1}^m$. In the off-policy setting, we require that the policy class $\widetilde{\Pi}$ contains all of these policies as well as the target policy—viz. $\{\mu_j\} \cup \{\pi\} \subseteq \widetilde{\Pi}$. Moreover, the predictor class $\widetilde{\mathcal{Q}}$ should contain the predictor class for the target policy, i.e., $\mathcal{Q}^\pi \subseteq \widetilde{\mathcal{Q}}$. We also assume weak realizability for this discussion.

Our result depends on a positive semidefinite matrix determined by the mixture weights $\{p_j\}_{j=1}^m$ along with the embeddings $\{\nu_{\mu_j}\}_{j=1}^m$ of the associated policies that generated the data. In particular, we define

$$\Sigma_\nu = \sum_{j=1}^{m} p_j^2 \nu_{\mu_j} \nu_{\mu_j}^\top.$$

Assuming that this is matrix is positive definite,[7] we define the norm $\|u\|_{\Sigma_\nu^{-1}} = \sqrt{u^T (\Sigma_\nu)^{-1} u}$. With this notation, we have the following bound.

**Lemma 6** (Concentrability with Bellman Rank). *For a mixture data-generation process and under the Bellman rank condition* (32)*, we have the upper bound*

$$K^\pi \leq \frac{1 + m\lambda}{1 + \lambda} \|\nu_\pi\|_{\Sigma_\nu^{-1}}^2, \tag{33}$$

*Proof.* Our proof exploits the upper bound (ii) from the claim (31) in Lemma 5. We first evaluate and redefine the ratio in this upper bound. Weak realizability coupled with the Bellman rank condition (32) implies that there exists some $Q_\star^\pi$ such that

$$0 = \langle f_j, \mathcal{B}^\pi Q_\star^\pi \rangle_\mu = p_j \mathbb{E}_{\mu_j} \mathcal{B}^\pi Q_\star^\pi = p_j \langle \nu_{\mu_j}, \xi_{Q_\star^\pi} \rangle, \qquad \text{for all } j = 1, \dots, m, \text{ and}$$
$$0 = \langle \mathbb{1}, \mathcal{B}^\pi Q_\star^\pi \rangle_\pi = \mathbb{E}_\pi \mathcal{B}^\pi Q_\star^\pi = \langle \nu_\pi, \xi_{Q_\star^\pi} \rangle.$$

---

[6]The original definition essentially takes $\widetilde{\Pi}$ as the set of all greedy policies with respect to $\widetilde{\mathcal{Q}}$. Since a dataset need not originate from greedy policies, the definition of Bellman rank is adapted in a natural way.

[7]If not, one can prove a result for a suitably regularized version.

Therefore, we have the equivalences $\mathbb{E}_{\mu_j}\mathcal{B}^\pi Q = \langle \nu_{\mu_j}, (\xi_Q - \xi_{Q^\pi_\star}) \rangle$ for all $j = 1, \ldots, m$, as well as $\mathbb{E}_\pi \mathcal{B}^\pi Q = \langle \nu_\pi, (\xi_Q - \xi_{Q^\pi_\star}) \rangle$. Introducing the shorthand $\Delta_Q = \xi_Q - \xi_{Q^\pi_\star}$, we can bound the ratio as follows

$$\sup_{Q \in \mathcal{C}^\pi_n} \left\{ \frac{(\langle \nu_\pi, \Delta_Q \rangle)^2}{\sum_{j=1}^m p_j^2 (\langle \nu_{\mu_j}, \Delta_Q \rangle)^2} \right\} = \sup_{Q \in \mathcal{C}^\pi_n} \left\{ \frac{(\langle \nu_\pi, \Delta_Q \rangle)^2}{\Delta_Q^\top \left( \sum_{j=1}^m p_j^2 \nu_{\mu_j} \nu_{\mu_j}^\top \right) \Delta_Q} \right\}$$

$$= \sup_{Q \in \mathcal{C}^\pi_n} \left\{ \frac{(\langle \nu_\pi, \Sigma_\nu^{-\frac{1}{2}} \widetilde{\Delta}_Q \rangle)^2}{\|\widetilde{\Delta}_Q\|_2^2} \right\} \qquad \text{where } \widetilde{\Delta}_Q = \Sigma_\nu^{\frac{1}{2}} \Delta_Q$$

$$\leq \|\nu_\pi\|_{\Sigma_\nu^{-1}}^2,$$

where the final step follows from the Cauchy–Schwarz inequality. $\qquad \square$

Thus, when performing off-policy evaluation with a mixture distribution under the Bellman rank condition, the coefficient $K^\pi$ is bounded by the alignment between the target policy $\pi$ and the data-generating distribution $\mu$, as measured in the the embedded space guaranteed by the Bellman rank condition. The structure of this upper bound is similar to a result that we derive in the sequel for linear approximation under Bellman closure (see Proposition 3).

### A.6   Further comments on the prediction error test space

A few comments on the bound in Lemma 1: as in our previous results, the pre-factor $\frac{\|\epsilon\|_\mu^2 + \lambda}{\|\mathbb{1}\|_\pi^2 + \lambda}$ serves as a normalization factor. Disregarding this leading term, the second ratio measures how the prediction error $\epsilon = Q - Q^\pi_\star$ along $\mu$ transfers to $\pi$, as measured via the operator $\mathcal{I} - \gamma \mathbb{P}^\pi$. This interaction is complex, since it includes the *bootstrapping term* $-\gamma \mathbb{P}^\pi$. (Notably, such a term is not present for standard prediction or bandit problems, in which case $\gamma = 0$.) This term reflects the dynamics intrinsic to reinforcement learning, and plays a key role in proving "hard" lower bounds for offline RL (e.g., see the work [Zan20]).

Observe that the bound in Lemma 1 requires only weak realizability, and thus it always applies. This fact is significant in light of a recent lower bound [FKSLX21], showing that without Bellman closure, off-policy learning is challenging even under strong concentrability assumption (such as bounds on density ratios). Lemma 1 gives a sufficient condition without Bellman closure, but with a different measure that accounts for bootstrapping.

If, in fact, (weak) Bellman closure holds, then Lemma 1 takes the following simplified form:

**Lemma 7** (OPC coefficient under Bellman closure). *If $\mathcal{E}^\pi \subseteq \mathcal{F}^\pi$ and weak Bellman closure holds, then*

$$K^\pi \leq \max_{\epsilon \in \mathcal{E}^\pi} \left\{ \frac{\|\epsilon\|_\mu^2 + \lambda}{1 + \lambda} \cdot \frac{\langle \mathbb{1}, \epsilon \rangle_\pi^2}{\langle \epsilon, \epsilon \rangle_\mu^2} \right\} \leq \max_{\epsilon \in \mathcal{E}^\pi} \left\{ \frac{\|\epsilon\|_\pi^2}{\|\epsilon\|_\mu^2} \right\}.$$

See Appendix D.3 for the proof.

In such case, the concentrability measures the increase in the discrepancy $Q - Q'$ of the feasible predictors when moving from the dataset distribution $\mu$ to the distribution of the target policy $\pi$. In Section 4.3, we give another bound under weak Bellman closure, and thereby recover a recent result due to Xie et al. [XCJ+21]. Finally, in Section 5, we provide some applications of this concentrability factor to the linear setting.

### A.7   From Importance Sampling to Bellman Closure

Let us show an application of Lemma 3 on an example with just two test spaces. Suppose that we suspect that Bellman closure holds, but rather than committing to such assumption, we wish to fall back to an importance sampling estimator if Bellman closure does not hold.

In order to streamline the presentation of the idea, let us introduce the following setup. Let $\pi^b$ be a behavioral policy that generates the dataset, i.e., such that each state-action $(s, a)$ in the dataset is

842 sampled from its discounted state distribution $d_{\pi^b}$. Next, let the identifier $o$ contain the trajectory
843 from $\nu_{\text{start}}$ up to the state-action pair $(s, a)$ recorded in the dataset. That is, each tuple $(s, a, r, s^+, o)$
844 in the dataset $\mathcal{D}$ is such that $(s, a) \sim d_{\pi^b}$ and $o$ contains the trajectory up to $(s, a)$.

845 We now define the test spaces. The first one is denoted with $\mathcal{F}_\pi^{\text{IS}}$ and leverages importance sampling.
846 It contains a single test function defined as the importance sampling estimator

$$\mathcal{F}_\pi^{\text{IS}} = \{f_\pi\}, \qquad \text{where } f_\pi(s, a, o) = \frac{1}{b_\pi} \prod_{(s_h, a_h) \in o} \frac{\pi(a_h \mid s_h)}{\pi^b(a_h \mid s_h)}. \tag{34}$$

847 The above product is over the random trajectory contained in the identifier $o$. The normalization
848 factor $b_\pi \in \mathbb{R}$ is connected to the maximum range of the importance sampling estimator, and ensures
849 that $\sup_{(s,a,o)} f_\pi(s, a, o) \leq 1$. The second test space is the prediction error test space $\mathcal{E}^\pi$ defined in
850 Section 4.2.

851 With this choice, let us define three concentrability coefficients. $K_{(1)}^\pi$ arises from importance sampling,
852 $K_{(2)}^\pi$ from the prediction error test space when Bellman closure holds and $K_{(3)}^\pi$ from the prediction
853 error test space when just weak realizability holds. They are defined as

$$K_{(1)}^\pi \leq \sqrt{b_\pi \frac{(1 + \lambda b_\pi)}{1 + \lambda}} \qquad K_{(2)}^\pi \leq \max_{\epsilon \in \mathcal{E}_\star^\pi} \frac{\langle \mathbb{1}, (\mathcal{I} - \gamma \mathbb{P}^\pi)\epsilon \rangle_\pi^2}{\langle \epsilon, (\mathcal{I} - \gamma \mathbb{P}^\pi)\epsilon \rangle_\mu^2} \times \frac{\|\epsilon\|_\mu^2 + \lambda}{\|\mathbb{1}\|_\pi^2 + \lambda}, \qquad K_{(3)}^\pi \leq c_1 \frac{\|\mathcal{B}^\pi Q\|_\pi^2}{\|\mathcal{B}^\pi Q\|_\mu^2}.$$

854 **Lemma 8** (From Importance Sampling to Bellman Closure)**.** *The choice* $\mathcal{F}^\pi = \mathcal{F}_\pi^{\text{IS}} \cup \mathcal{E}^\pi$ *for all* $\pi \in$
855 $\Pi$ *ensures that with probability at least* $1 - \delta$*, the oracle inequality* (9) *holds with* $K^\pi \leq$
856 $\min\{K_{(1)}^\pi, K_{(2)}^\pi, K_{(3)}^\pi\}$ *if weak Bellman closure holds and* $K^\pi \leq \min\{K_{(1)}^\pi, K_{(2)}^\pi\}$ *otherwise.*

857 *Proof.* Let us calculate the off-policy cost coefficient associated with $\mathcal{F}_\pi^{\text{IS}}$. The unbiasedness of the
858 importance sampling estimator gives us the following population constraint (here $\mu = d_{\pi^b}$)

$$|\langle f_\pi, \mathcal{B}^\pi Q \rangle_\mu| = |\mathbb{E}_\mu f_\pi \mathcal{B}^\pi Q| = \frac{1}{b_\pi} |\mathbb{E}_\pi \mathcal{B}^\pi Q| = \frac{1}{b_\pi} |\langle \mathbb{1}, \mathcal{B}^\pi Q \rangle_\pi| \leq \frac{L}{\sqrt{n}} \sqrt{\|f_\pi\|_2^2 + \lambda}$$

859 The norm of the test function reads (notice that $\mu$ generates $(s, a, o)$ here)

$$\|f_\pi\|_\mu^2 = \mathbb{E}_\mu f_\pi^2 = \frac{1}{b_\pi^2} \mathbb{E}_\mu \left[ \prod_{(s_h, a_h) \in o} \frac{\pi(a_h \mid s_h)}{\pi^b(a_h \mid s_h)} \right]^2 = \frac{1}{b_\pi^2} \mathbb{E}_\pi \left[ \prod_{(s_h, a_h) \in o} \frac{\pi(a_h \mid s_h)}{\pi^b(a_h \mid s_h)} \right] \leq \frac{1}{b_\pi}.$$

860 Together with the prior display, we obtain

$$\frac{\langle \mathbb{1}, \mathcal{B}^\pi Q \rangle_\pi^2}{b_\pi^2 (\|f_\pi\|_2^2 + \lambda)} \leq \frac{\rho}{n}.$$

861 The resulting concentrability coefficient is therefore

$$K^\pi \leq \max_{Q \in \mathcal{C}_n^\pi} \frac{\langle \mathbb{1}, \mathcal{B}^\pi Q \rangle_\pi^2}{1 + \lambda} \times \frac{n}{\rho} \leq \max_{Q \in \mathcal{C}_n^\pi} \frac{\langle \mathbb{1}, \mathcal{B}^\pi Q \rangle_\pi^2}{1 + \lambda} \times \frac{b_\pi^2 (\|f_\pi\|_2^2 + \lambda)}{\langle \mathbb{1}, \mathcal{B}^\pi Q \rangle_\pi^2} \leq b_\pi \frac{(1 + \lambda b_\pi)}{1 + \lambda}.$$

862 Chaining the above result with Lemmas 1 and 2, using Lemma 3 and plugging back into Theorem 3
863 yields the thesis. $\qquad \square$

## A.8 Implementation for Off-Policy Predictions

865 In this section, we describe a computationally efficient way in which to compute the upper/lower
866 estimates (5). Given a finite set of $n_\mathcal{F}$ test functions, it involves solving a quadratic program with
867 $2n_\mathcal{F} + 1$ constraints.

868 Let us first work out a concise description of the constraints defining membership in $\widehat{\mathcal{C}}_n^\pi$. Introduce the
869 shorthand $n_f \overset{def}{=} \|f_j\|_n^2 + \lambda$. We then define the empirical average feature vector $\widehat{\phi}_f$, the empirical

average reward $\widehat{r}_f$, and the average next-state feature vector $\widehat{\phi}_f^{+\pi}$ as

$$\widehat{\phi}_f = \frac{1}{\sqrt{n_f}} \sum_{(s,a,r,s^+)\in\mathcal{D}} f(s,a)\phi(s,a), \qquad \widehat{r}_f = \frac{1}{\sqrt{n_f}} \sum_{(s,a,r,s^+)\in\mathcal{D}} f(s,a)r,$$

$$\widehat{\phi}_f^{+\pi} = \frac{1}{\sqrt{n_f}} \sum_{(s,a,r,s^+)\in\mathcal{D}} f(s,a)\phi(s^+,\pi).$$

In terms of this notation, each empirical constraint defining $\widehat{\mathcal{C}}_n^\pi$ can be written in the more compact form

$$\frac{|\langle f, \delta^\pi Q\rangle_n|}{\sqrt{n_f}} = \left| \langle \widehat{\phi}_f - \gamma\widehat{\phi}_f^{+\pi},\, w\rangle - \widehat{r}_f \right| \le \sqrt{\frac{\rho}{n}}.$$

Then the set of empirical constraints can be written as a set of constraints linear in the critic parameter $w$ coupled with the assumed regularity bound on $w$

$$\widehat{\mathcal{C}}_n^\pi = \left\{ w \in \mathbb{R}^d \mid \|w\|_2 \le 1, \quad \text{and} \quad -\sqrt{\frac{\rho}{n}} \le \langle \widehat{\phi}_f - \gamma\widehat{\phi}_f^{+\pi},\, w\rangle - \widehat{r}_f \le \sqrt{\frac{\rho}{n}} \quad \text{for all } f \in \mathcal{F}^\pi \right\}.$$
$$(35)$$

Thus, the estimates $\widehat{V}_{\min}^\pi$ (respectively $\widehat{V}_{\max}^\pi$) acan be computed by minimizing (respectively maximizing) the linear objective function $w \mapsto \langle [\mathbb{E}_{s\sim\nu_{\text{start}}}\mathbb{E}_{a\sim\pi}\phi(s,a)],\, w\rangle$ subject to the $2n_{\mathcal{F}} + 1$ constraints in equation (35). Therefore, the estimates can be computed in polynomial time for any test function with a cardinality that grows polynomially in the problem parameters.

## A.9 Discussion of Linear Approximate Optimization

Here we discuss the presence of the supremum over policies in the coefficient $K_{(1)}^{\widetilde{\pi}}$ from equation (23). In particular, it arises because our actor-critic method iteratively approximates the maximum in the max-min estimate (6) using a gradient-based scheme. The ability of a gradient-based method to make progress is related to the estimation accuracy of the gradient, which is the $Q$ estimates of the actor's current policy $\pi_t$; more specifically, the gradient is the $Q$ function parameter $w_t$. In the general case, the estimation error of the gradient $w_t$ depends on the policy under consideration through the matrix $\Sigma_{\lambda,\text{Boot}}^{+\pi_t}$, while it is independent in the special case of Bellman closure (as it depends on just $\Sigma$). As the actor's policies are random, this yields the introduction of a $\sup_{\pi\in\Pi}$ in the general bound. Notice the method still competes with the best comparator $\widetilde{\pi}$ by measuring the errors along the distribution of the comparator (through the operator $\mathbb{E}_{\widetilde{\pi}}$). To be clear, $\sup_{\pi\in\Pi}$ may not arise with approximate solution methods that do not rely only on the gradient to make progress (such as second-order methods); we leave this for future research. Reassuringly, when Bellman closure, the approximate solution method recovers the standard guarantees established in the paper [ZWB21].

## B General Guarantees

### B.1 A deterministic guarantee

We begin our analysis stating a deterministic set of sufficient conditions for our estimators to satisfy the guarantees (8) and (9). This formulation is useful, because it reveals the structural conditions that underlie success of our estimators, and in particular the connection to weak realizability. In Section B.2, we exploit this deterministic result to show that, under a fairly general sampling model, our estimators enjoy these guarantees with high probability.

In the previous section, we introduced the population level set $\mathcal{C}_n^\pi$ that arises in the statement of our guarantees. Also central in our analysis is the infinite data limit of this set. More specifically, for any fixed $(\rho, \lambda)$, if we take the limit $n \to \infty$, then $\mathcal{C}_n^\pi$ reduces to the set of all solutions to the weak formulation (25)—that is

$$\mathcal{C}_\infty^\pi(\mathcal{F}^\pi) = \{Q \in \mathcal{Q}^\pi \mid \langle f, \mathcal{B}^\pi Q \rangle_\mu = 0 \quad \text{for all } f \in \mathcal{F}^\pi\}. \tag{36}$$

As before, we omit the dependence on the test function class $\mathcal{F}^\pi$ when it is clear from context. By construction, we have the inclusion $\mathcal{C}_\infty^\pi(\mathcal{F}^\pi) \subseteq \mathcal{C}_n^\pi(4\rho, \lambda; \mathcal{F}^\pi)$ for any non-negative pair $(\rho, \lambda)$.

Our first set of guarantees hold when the random set $\widehat{\mathcal{C}}_n^\pi$ satisfies the *sandwich relation*

$$\mathcal{C}_\infty^\pi(\mathcal{F}^\pi) \subseteq \widehat{\mathcal{C}}_n^\pi(\rho, \lambda; \mathcal{F}^\pi) \subseteq \mathcal{C}_n^\pi(4\rho, \lambda; \mathcal{F}^\pi) \tag{37}$$

To provide intuition as to why this sandwich condition is natural, observe that it has two important implications:

   (a) Recalling the definition of weak realizability (1), the weak solution $Q_\star^\pi$ belongs to the empirical constraint set $\widehat{\mathcal{C}}_n^\pi$ for any choice of test function space. This important property follows because $Q_\star^\pi$ must satisfy the constraints (25), and thus it belongs to $\mathcal{C}_\infty^\pi \subseteq \widehat{\mathcal{C}}_n^\pi$.

   (b) All solutions in $\widehat{\mathcal{C}}_n^\pi$ also belong to $\mathcal{C}_n^\pi$, which means they approximately satisfy the weak Bellman equations in a way quantified by $\mathcal{C}_n^\pi$.

By leveraging these facts in the appropriate way, we can establish the following guarantee:

**Proposition 4.** *The following two statements hold.*

   (a) *Policy evaluation: If the set $\widehat{\mathcal{C}}_n^\pi$ satisfies the sandwich relation (37), then the estimates $(\widehat{V}_{min}^\pi, \widehat{V}_{max}^\pi)$ satisfy the width bound (8b). If, in addition, weak Bellman realizability for $\pi$ is assumed, then the coverage (8a) condition holds.*

   (b) *Policy optimization: If the sandwich relation (37) and weak Bellman realizability hold for all $\pi \in \Pi$, then any max-min (6) optimal policy $\widetilde{\pi}$ satisfies the oracle inequality (9).*

See Section C.1 for the proof of this claim.

In summary, Proposition 4 ensures that when weak realizability is in force, then the sandwich relation (37) is a sufficient condition for both the policy evaluation (8) and optimization (9) guarantees to hold. Accordingly, the next phase of our analysis focuses on deriving sufficient conditions for the sandwich relation to hold with high probabability.

### B.2 Some high-probability guarantees

As stated, Proposition 4 is a "meta-result", in that it applies to any choice of set $\widehat{\mathcal{C}}_n^\pi \equiv \widehat{\mathcal{C}}_n^\pi(\rho, \lambda; \mathcal{F}^\pi)$ for which the sandwich relation (37) holds. In order to obtain a more concrete guarantee, we need to impose assumptions on the way in which the dataset was generated, and concrete choices of $(\rho, \lambda)$ that suffice to ensure that the associated sandwich relation (37) holds with high probability. These tasks are the focus of this section.

### B.2.1 A model for data generation

Let us begin by describing a fairly general model for data-generation. Any sample takes the form $z \overset{def}{=} (s, a, r, s^+, o)$, where the five components are defined as follows:

- the pair $(s, a)$ index the current state and action.

- the random variable $r$ is a noisy observation of the mean reward.

- the random state $s^+$ is the next-state sample, drawn according to the transition $\mathbb{P}(s, a)$.

- the variable $o$ is an optional identifier.

As one example of the use of an identifier variable, if samples might be generated by one of two possible policies—say $\pi_1$ and $\pi_2$—the identifier can take values in the set $\{1, 2\}$ to indicate which policy was used for a particular sample.

Overall, we observe a dataset $\mathcal{D} = \{z_i\}_{i=1}^n$ of $n$ such quintuples. In the simplest of possible settings, each triple $(s, a, o)$ is drawn i.i.d. from some fixed distribution $\mu$, and the noisy reward $r_i$ is an unbiased estimate of the mean reward function $R(s_i, a_i)$. In this case, our dataset consists of $n$ i.i.d. quintuples. More generally, we would like to accommodate richer sampling models in which the sample $z_i = (s_i, a_i, o_i, r_i, s_i^+)$ at a given time $i$ is allowed to depend on past samples. In order to specify such dependence in a precise way, define the nested sequence of sigma-fields

$$\mathcal{F}_1 = \emptyset, \quad \text{and} \quad \mathcal{F}_i \overset{def}{=} \sigma\Big(\{z_j\}_{j=1}^{i-1}\Big) \qquad \text{for } i = 2, \ldots, n. \tag{38}$$

In terms of this filtration, we make the following definition:

**Assumption 3** (Adapted dataset). *An adapted dataset is a collection $\mathcal{D} = \{z_i\}_{i=1}^n$ such that for each $i = 1, \ldots, n$:*

- *There is a conditional distribution $\mu_i$ such that $(s_i, a_i, o_i) \sim \mu_i(\cdot \mid \mathcal{F}_i)$.*

- *Conditioned on $(s_i, a_i, o_i)$, we observe a noisy reward $r_i = r(s_i, a_i) + \eta_i$ with $\mathbb{E}[\eta_i \mid \mathcal{F}_i] = 0$, and $|r_i| \leq 1$.*

- *Conditioned on $(s_i, a_i, o_i)$, the next state $s_i^+$ is generated according to $\mathbb{P}(s_i, a_i)$.*

Under this assumption, we can define the (possibly) random reference measure

$$\mu(s, a, o) \overset{def}{=} \frac{1}{n} \sum_{i=1}^n \mu_i\big(s, a, o \mid \mathcal{F}_i\big). \tag{39}$$

In words, it corresponds to the distribution induced by first drawing a time index $i \in \{1, \ldots, n\}$ uniformly at random, and then sampling a triple $(s, a, o)$ from the conditional distribution $\mu_i(\cdot \mid \mathcal{F}_i)$.

### B.2.2 A general guarantee

Recall that there are three function classes that underlie our method: the test function class $\mathcal{F}$, the policy class $\Pi$, and the $Q$-function class $\mathcal{Q}$. In this section, we state a general guarantee (Theorem 3) that involves the metric entropies of these sets. In Section B.2.3, we provide corollaries of this guarantee for specific function classes.

In more detail, we equip the test function class and the $Q$-function class with the usual sup-norm

$$\|f - \tilde{f}\|_\infty \overset{def}{=} \sup_{(s,a,o)} |f(s, a, o) - \tilde{f}(s, a, o)|, \quad \text{and} \quad \|Q - \tilde{Q}\|_\infty \overset{def}{=} \sup_{(s,a)} |Q(s, a) - \tilde{Q}(s, a)|,$$

and the policy class with the sup-TV norm

$$\|\pi - \tilde{\pi}\|_{\infty,1} \overset{def}{=} \sup_s \|\pi(\cdot \mid s) - \tilde{\pi}(\cdot \mid s)\|_1 = \sup_s \sum_a |\pi(a \mid s) - \tilde{\pi}(a \mid s)|.$$

For a given $\epsilon > 0$, we let $\mathcal{N}_\epsilon(\mathcal{F})$, $\mathcal{N}_\epsilon(\mathcal{Q})$, and $\mathcal{N}_\epsilon(\Pi)$ denote the $\epsilon$-covering numbers of each of these function classes in the given norms. Given these covering numbers, a tolerance parameter $\delta \in (0, 1)$

and the shorthand $\phi(t) = \max\{t, \sqrt{t}\}$, define the radius function

$$\rho(\epsilon, \delta) \stackrel{def}{=} n\Big\{ \int_{\epsilon^2}^{\epsilon} \phi\Big(\frac{\log N_u(\mathcal{F})}{n}\Big) du + \frac{\log N_\epsilon(\mathcal{Q})}{n} + \frac{\log N_\epsilon(\Pi)}{n} + \frac{\log(n/\delta)}{n}\Big\}. \qquad (40a)$$

In our theorem, we implement the estimator using a radius $\rho = \rho(\epsilon, \delta)$, where $\epsilon > 0$ is any parameter that satisfies the bound

$$\epsilon^2 \stackrel{(i)}{\leq} \bar{c}\,\frac{\rho(\epsilon, \delta)}{n}, \quad \text{and} \quad \lambda \stackrel{(i)}{=} 4\frac{\rho(\epsilon, \delta)}{n}. \qquad (40b)$$

Here $\bar{c} > 0$ is a suitably chosen but universal constant (whose value is determined in the proof), and we adopt the shorthand $\rho = \rho(\epsilon, \delta)$ in our statement below.

**Theorem 3** (High-probability guarantees)**.** *Consider the estimates implemented using triple $(\Pi, \mathcal{F}, \mathcal{Q})$ that is weakly Bellman realizable (Assumption 1); an adapted dataset (Assumption 3); and with the choices (40) for $(\epsilon, \rho, \lambda)$. Then with probability at least $1 - \delta$:*

Policy evaluation: *For any $\pi \in \Pi$, the estimates $(\widehat{V}_{min}^\pi, \widehat{V}_{max}^\pi)$ specify a confidence interval satisfying the coverage (8a) and width bounds (8b).*

Policy optimization: *Any max-min policy (6) $\widetilde{\pi}$ satisfies the oracle inequality (9).*

See Appendix C.3 for the proof of the claim.

**Choices of** $(\rho, \epsilon, \lambda)$**:**  Let us provide a few comments about the choices of $(\rho, \epsilon, \lambda)$ from equations (40a) and (40b). The quality of our bounds depends on the size of the constraint set $\mathcal{C}_n^\pi$, which is controlled by the constraint level $\sqrt{\frac{\rho}{n}}$. Consequently, our results are tightest when $\rho = \rho(\epsilon, \delta)$ is as small as possible. Note that $\rho$ is an decreasing function of $\epsilon$, so that in order to minimize it, we would like to choose $\epsilon$ as large as possible subject to the constraint (40b)(i). Ignoring the entropy integral term in equation (40b) for the moment—see below for some comments on it—these considerations lead to

$$n\epsilon^2 \asymp \log N_\epsilon(\mathcal{F}) + \log N_\epsilon(\mathcal{Q}) + \log N_\epsilon(\Pi). \qquad (41)$$

This type of relation for the choice of $\epsilon$ in non-parametric statistics is well-known (e.g., see Chapters 13–15 in the book [Wai19] and references therein). Moreover, setting $\lambda \asymp \epsilon^2$ as in equation (40b)(ii) is often the correct scale of regularization.

**Key technical steps in proof:**  It is worthwhile making a few comments about the structure of the proof so as to clarify the connections to Proposition 4 along with the weak formulation that underlies our methods. Recall that Proposition 4 requires the empirical $\widehat{\mathcal{C}}_n^\pi$ and population sets $\mathcal{C}_n^\pi$ to satisfy the sandwich relation (37). In order to prove that this condition holds with high probability, we need to establish uniform control over the family of random variables

$$\frac{\big| \langle f, \delta^\pi(Q) \rangle_n - \langle f, \mathcal{B}^\pi(Q) \rangle_\mu \big|}{\sqrt{\|f\|_n^2 + \lambda}}, \qquad \text{as indexed by the triple } (f, Q, \pi). \qquad (42)$$

Note that the differences in the numerator of these variables correspond to moving from the empirical constraints on $Q$-functions that are enforced using the TD errors, to the population constraints that involve the Bellman error function.

Uniform control of the family (42), along with the differences $\|f\|_n - \|f\|_\mu$ uniformly over $f$, allows us to relate the empirical and population sets, since the associated constraints are obtained by shifting between the empirical inner products $\langle \cdot, \cdot \rangle_n$ to the reference inner products $\langle \cdot, \cdot \rangle_\mu$. A simple discretization argument allows us to control the differences uniformly in $(Q, \pi)$, as reflected by the metric entropies appearing in our definition (40). Deriving uniform bounds over test functions $f$—due to the self-normalizing nature of the constraints—requires a more delicate argument. More precisely, in order to obtain optimal results for non-parametric problems (see Corollary 2 to follow), we need to localize the empirical process at a scale $\epsilon$, and derive bounds on the localized increments. This portion of the argument leads to the entropy integral—which is localized to the interval $[\epsilon^2, \epsilon]$—in our definition (40a) of the radius function.

**Intuition from the on-policy setting:** In order to gain intuition for the statistical meaning of the guarantees in Theorem 3, it is worthwhile understanding the implications in a rather special case— namely, the simpler on-policy setting, where the discounted occupation measure induced by the target policy $\pi$ coincides with the dataset distribution $\mu$. Let us consider the case in which the identity function $\mathbb{1}$ belongs to the test class $\mathcal{F}^\pi$. Under these conditions, for any $Q \in \mathcal{C}_n^\pi$, we can write

$$\max_{Q \in \mathcal{C}_n^\pi} |\mathbb{E}_\pi \mathcal{B}^\pi Q| \overset{(i)}{=} \max_{Q \in \mathcal{C}_n^\pi} |\mathbb{E}_\mu \mathcal{B}^\pi Q| \overset{(ii)}{\leq} \sqrt{1+\lambda} \sqrt{\frac{\rho}{n}},$$

where equality (i) follows from the on-policy assumption, and step (ii) follows from the definition of the set $\mathcal{C}_n^\pi$, along with the condition that $\mathbb{1} \in \mathcal{F}^\pi$. Consequently, in the on-policy setting, the width bound (8b) ensures that

$$|\widehat{V}_{\min}^\pi - \widehat{V}_{\max}^\pi| \leq 2\frac{\sqrt{1+\lambda}}{1-\gamma}\sqrt{\frac{\rho}{n}}. \tag{43}$$

In this simple case, we see that the confidence interval scales as $\sqrt{\rho/n}$, where the quantity $\rho$ is related to the metric entropy via equation (40b). In the more general off-policy setting, the bound involves this term, along with additional terms that reflect the cost of off-policy data. We discuss these issues in more detail in Section 4. Before doing so, however, it is useful derive some specific corollaries that show the form of $\rho$ under particular assumptions on the underlying function classes, which we now do.

### B.2.3 Some corollaries

Theorem 3 applies generally to triples of function classes $(\Pi, \mathcal{F}, \mathcal{Q})$, and the statistical error $\sqrt{\frac{\rho(\epsilon, \delta)}{n}}$ depends on the metric entropies of these function classes via the definition (40a) of $\rho(\epsilon, \delta)$, and the choices (40b). As shown in this section, if we make particular assumptions about the metric entropies, then we can derive more concrete guarantees.

**Parametric and finite VC classes:** One form of metric entropy, typical for a relatively simple function class $\mathcal{G}$ (such as those with finite VC dimension) scales as

$$\log N_\epsilon(\mathcal{G}) \asymp d \, \log\left(\tfrac{1}{\epsilon}\right), \tag{44}$$

for some dimensionality parameter $d$. For instance, bounds of this type hold for linear function classes with $d$ parameters, and for finite VC classes (with $d$ proportional to the VC dimension); see Chapter 5 of the book [Wai19] for more details.

**Corollary 1.** *Suppose each class of the triple $(\Pi, \mathcal{F}, \mathcal{Q})$ has metric entropy that is at most polynomial (44) of order $d$. Then for a sample size $n \geq 2d$, the claims of Theorem 3 hold with $\epsilon^2 = d/n$ and*

$$\tilde{\rho}\left(\sqrt{\tfrac{d}{n}}, \delta\right) \overset{def}{=} c \left\{ d \, \log\left(\tfrac{n}{d}\right) + \log\left(\tfrac{n}{\delta}\right) \right\}, \tag{45}$$

*where $c$ is a universal constant.*

*Proof.* Our strategy is to upper bound the radius $\rho$ from equation (40a), and then show that this upper bound $\tilde{\rho}$ satisfies the conditions (40b) for the specified choice of $\epsilon^2$. We first control the term $\log N_\epsilon(\mathcal{F})$. We have

$$\frac{1}{\sqrt{n}} \int_{\epsilon^2}^\epsilon \sqrt{\log N_u(\mathcal{F})} du \leq \sqrt{\frac{d}{n}} \int_0^\epsilon \sqrt{\log(1/u)} du \;=\; \epsilon\sqrt{\frac{d}{n}} \int_0^1 \sqrt{\log(1/(\epsilon t))} dt \;=\; c\epsilon \log(1/\epsilon)\sqrt{\frac{d}{n}}.$$

Similarly, we have

$$\frac{1}{n} \int_{\epsilon^2}^\epsilon \log N_u(\mathcal{F}) du \leq \epsilon\frac{d}{n}\left\{ \int_\epsilon^1 \log(1/t) dt + \log(1/\epsilon) \right\} \;\leq\; c\,\epsilon \log(1/\epsilon)\frac{d}{n}.$$

Finally, for terms not involving entropy integrals, we have

$$\max\left\{ \frac{\log N_\epsilon(\mathcal{Q})}{n}, \frac{\log N_\epsilon(\Pi)}{n} \right\} \leq c\frac{d}{n} \log(1/\epsilon).$$

Setting $\epsilon^2 = d/n$, we see that the required conditions (40b) hold with the specified choice (45) of $\tilde{\rho}$. $\square$

**Richer function classes:** In the previous section, the metric entropy scaled logarithmically in the inverse precision $1/\epsilon$. For other (richer) function classes, the metric entropy exhibits a polynomial scaling in the inverse precision, with an exponent $\alpha > 0$ that controls the complexity. More precisely, we consider classes of the form

$$\log N_\epsilon(\mathcal{G}) \asymp \left(\frac{1}{\epsilon}\right)^\alpha. \tag{46}$$

For example, the class of Lipschitz functions in dimension $d$ has this type of metric entropy with $\alpha = d$. More generally, for Sobolev spaces of functions that have $s$ derivatives (and the $s^{th}$-derivative is Lipschitz), we encounter metric entropies of this type with $\alpha = d/s$. See Chapter 5 of the book [Wai19] for further background.

**Corollary 2.** *Suppose that each function class $(\Pi, \mathcal{F}, \mathcal{Q})$ has metric entropy with at most $\alpha$-scaling (46) for some $\alpha \in (0, 2)$. Then the claims of Theorem 3 hold with $\epsilon^2 = (1/n)^{\frac{2}{2+\alpha}}$, and*

$$\tilde{\rho}\left((1/n)^{\frac{1}{2+\alpha}}, \delta\right) = c \left\{ n^{\frac{\alpha}{2+\alpha}} + \log(n/\delta) \right\}. \tag{47}$$

*where $c$ is a universal constant.*

We note that for standard regression problems over classes with $\alpha$-metric entropy, the rate $(1/n)^{\frac{2}{2+\alpha}}$ is well-known to be minimax optimal (e.g., see Chapter 15 in the book [Wai19], as well as references therein).

*Proof.* We start by controlling the terms involving entropy integrals. In particular, we have

$$\frac{1}{\sqrt{n}} \int_{\epsilon^2}^{\epsilon} \sqrt{\log N_u(\mathcal{F})} du \leq \frac{c}{\sqrt{n}} u^{1-\frac{\alpha}{2}} \Big|_0^{\epsilon} = \frac{c}{\sqrt{n}} \epsilon^{1-\frac{\alpha}{2}}.$$

Requiring that this term is of order $\epsilon^2$ amounts to enforcing that $\epsilon^{1+\frac{\alpha}{2}} \asymp (1/\sqrt{n})$, or equivalently that $\epsilon^2 \asymp (1/n)^{\frac{2}{2+\alpha}}$.

If $\alpha \in (0, 1]$, then the second entropy integral converges and is of lower order. Otherwise, if $\alpha \in (1, 2)$, then we have

$$\frac{1}{n} \int_{\epsilon^2}^{\epsilon} \log N_u(\mathcal{F}) du \leq \frac{c}{n} \int_{\epsilon^2}^{\epsilon} (1/u)^\alpha du \leq \frac{c}{n} (\epsilon^2)^{1-\alpha}.$$

Hence the requirement that this term is bounded by $\epsilon^2$ is equivalent to $\epsilon^{2\alpha} \gtrsim (1/n)$, or $\epsilon^2 \gtrsim (1/n)^{1/\alpha}$. When $\alpha \in (1, 2)$, we have $\frac{1}{\alpha} > \frac{2}{2+\alpha}$, so that this condition is milder than our first condition.

Finally, we have $\max \left\{ \frac{\log N_\epsilon(\mathcal{Q})}{n}, \frac{\log N_\epsilon(\Pi)}{n} \right\} \leq \frac{c}{n} (1/\epsilon)^\alpha$, and requiring that this term scales as $\epsilon^2$ amounts to requiring that $\epsilon^{2+\alpha} \asymp (1/n)$, or equivalently $\epsilon^2 \asymp (1/n)^{\frac{2}{2+\alpha}}$, as before. $\square$

 # C  Main Proofs

1071 This section is devoted to the proofs of our guarantees for general function classes—namely, Proposi-
1072 tion 4 that holds in a deterministic manner, and Theorem 3 that gives high probability bounds under a
1073 particular sampling model.

## C.1  Proof of Proposition 4

1075 Our proof makes use of an elementary simulation lemma, which we state here:

1076 **Lemma 9** (Simulation lemma). *For any policy $\pi$ and function $Q$, we have*

$$\mathbb{E}_{S \sim \nu_{\text{start}}}(Q - Q^\pi)(S, \pi) = \frac{\mathbb{E}_\pi \mathcal{B}^\pi Q}{1 - \gamma} \tag{48}$$

1077 See Appendix C.2 for the proof of this claim.

### C.1.1  Proof of policy evaluation claims

1079 First of all, we have the elementary bounds

$$|\widehat{V}_{\min}^\pi - V^\pi| = |\min_{Q \in \widehat{\mathcal{C}}_n^\pi} \mathbb{E}_{S \sim \nu_{\text{start}}} Q(S, \pi) - V^\pi| \leq \max_{Q \in \widehat{\mathcal{C}}_n^\pi} |\mathbb{E}_{S \sim \nu_{\text{start}}} Q(S, \pi) - V^\pi|, \quad \text{and}$$

$$|\widehat{V}_{\max}^\pi - V^\pi| = |\max_{Q \in \widehat{\mathcal{C}}_n^\pi} \mathbb{E}_{S \sim \nu_{\text{start}}} Q(S, \pi) - V^\pi| \leq \max_{Q \in \widehat{\mathcal{C}}_n^\pi} |\mathbb{E}_{S \sim \nu_{\text{start}}} Q(S, \pi) - V^\pi|.$$

1080 Consequently, in order to prove the bound (8b) it suffices to upper bound the right-hand side common
1081 in the two above displays. Since $\widehat{\mathcal{C}}_n^\pi \subseteq \mathcal{C}_n^\pi$, we have the upper bound

$$\max_{Q \in \widehat{\mathcal{C}}_n^\pi} |\mathbb{E}_{S \sim \nu_{\text{start}}} Q(S, \pi) - V^\pi| \leq \max_{Q \in \mathcal{C}_n^\pi} |\mathbb{E}_{S \sim \nu_{\text{start}}} Q(S, \pi) - V^\pi|$$

$$= \max_{Q \in \mathcal{C}_n^\pi} |\mathbb{E}_{S \sim \nu_{\text{start}}} [Q(S, \pi) - Q^\pi(S, \pi)]|$$

$$\overset{\text{(i)}}{=} \frac{1}{1 - \gamma} \max_{Q \in \mathcal{C}_n^\pi} \frac{\mathbb{E}_\pi \mathcal{B}^\pi Q}{1 - \gamma}$$

1082 where step (i) follows from Lemma 9. Combined with the earlier displays, this completes the proof
1083 of the bound (8b).

1084 We now show the inclusion $[\widehat{V}_{\min}^\pi, \widehat{V}_{\max}^\pi] \ni V^\pi$ when weak realizability holds. By definition of weak
1085 realizability, there exists some $Q_\star^\pi \in \mathcal{C}_\infty^\pi$. In conjunction with our sandwich assumption, we are
1086 guaranteed that $Q_\star^\pi \in \mathcal{C}_\infty^\pi \subseteq \widehat{\mathcal{C}}_n^\pi$, and consequently

$$\widehat{V}_{\min}^\pi = \min_{Q \in \widehat{\mathcal{C}}_n^\pi} \mathbb{E}_{S \sim \nu_{\text{start}}} Q(S, \pi) \leq \min_{Q \in \mathcal{C}_\infty^\pi} \mathbb{E}_{S \sim \nu_{\text{start}}} Q(S, \pi) \leq \mathbb{E}_{S \sim \nu_{\text{start}}} Q_\star^\pi(S, \pi) = V^\pi, \quad \text{and}$$

$$\widehat{V}_{\max}^\pi = \max_{Q \in \widehat{\mathcal{C}}_n^\pi} \mathbb{E}_{S \sim \nu_{\text{start}}} Q(S, \pi) \geq \max_{Q \in \mathcal{C}_\infty^\pi} \mathbb{E}_{S \sim \nu_{\text{start}}} Q(S, \pi) \geq \mathbb{E}_{S \sim \nu_{\text{start}}} Q_\star^\pi(S, \pi) = V^\pi.$$

### C.1.2  Proof of policy optimization claims

1088 We now prove the oracle inequality (9) on the value $V^{\widetilde{\pi}}$ of a policy $\widetilde{\pi}$ that optimizes the max-min
1089 criterion. Fix an arbitrary comparator policy $\pi$. Starting with the inclusion $[\widehat{V}_{\min}^{\widetilde{\pi}}, \widehat{V}_{\max}^{\widetilde{\pi}}] \ni V^{\widetilde{\pi}}$, we
1090 have

$$V^{\widetilde{\pi}} \overset{(i)}{\geq} \widehat{V}_{\min}^{\widetilde{\pi}} \overset{(ii)}{\geq} \widehat{V}_{\min}^\pi = V^\pi - \left(V^\pi - \widehat{V}_{\min}^\pi\right) \overset{(iii)}{\geq} V^\pi - \frac{1}{1 - \gamma} \max_{Q \in \mathcal{C}_n^\pi} \frac{|\mathbb{E}_\pi \mathcal{B}^\pi Q|}{1 - \gamma},$$

1091 where step (i) follows from the stated inclusion at the start of the argument; step (ii) follows
1092 since $\widetilde{\pi}$ solves the max-min program; and step (iii) follows from the bound $|V^\pi - \widehat{V}_{\min}^\pi| \leq$
1093 $\frac{1}{1-\gamma} \max_{Q \in \mathcal{C}_n^\pi} \frac{\mathbb{E}_\pi \mathcal{B}^\pi Q}{1-\gamma}$, as proved in the preceding section. This lower bound holds uniformly
1094 for all comparators $\pi$, from which the stated claim follows.

## C.2 Proof of Lemma 9

For each $t = 1, 2, \ldots$, let $\mathbb{E}_t$ be the expectation over the state-action pair at timestep $t$ upon starting from $\nu_{\text{start}}$, so that we have $\mathbb{E}_{S \sim \nu_{\text{start}}}(Q - Q^\pi)(S, \pi) = \mathbb{E}_0[Q - Q^\pi]$ by definition. We claim that

$$\mathbb{E}_0[Q - Q^\pi] = \sum_{\tau=1}^t \gamma^{\tau-1} \mathbb{E}_{\tau-1} \mathcal{B}^\pi Q + \gamma^t \mathbb{E}_t[Q - Q^\pi] \qquad \text{for all } t = 1, 2, \ldots. \tag{49}$$

For the base case $t = 1$, we have

$$\mathbb{E}_0[Q - Q^\pi] = \mathbb{E}_0[Q - \mathcal{T}^\pi Q] + \mathbb{E}_0[\mathcal{T}^\pi Q - \mathcal{T}^\pi Q^\pi] = \mathbb{E}_0[Q - \mathcal{T}^\pi Q] + \gamma \mathbb{E}_1[Q - Q^\pi], \tag{50}$$

where we have used the definition of the Bellman evaluation operator to assert that $\mathbb{E}_0[\mathcal{T}^\pi Q - \mathcal{T}^\pi Q^\pi] = \gamma \mathbb{E}_1[Q - Q^\pi]$. Since $Q - \mathcal{T}^\pi Q = \mathcal{B}^\pi Q$, the equality (50) is equivalent to the claim (49) with $t = 1$.

Turning to the induction step, we now assume that the claim (49) holds for some $t \geq 1$, and show that it holds at step $t + 1$. By a similar argument, we can write

$$\gamma^t \mathbb{E}_t[Q - Q^\pi] = \gamma^t \mathbb{E}_t[Q - \mathcal{T}^\pi Q + \mathcal{T}^\pi Q - \mathcal{T}^\pi Q^\pi] = \gamma^t \mathbb{E}_t[Q - \mathcal{T}^\pi Q] + \gamma^{t+1} \mathbb{E}_{t+1}[Q - Q^\pi]$$
$$= \gamma^t \mathbb{E}_t \mathcal{B}^\pi Q + \gamma^{t+1} \mathbb{E}_{t+1}[Q - Q^\pi].$$

By the induction hypothesis, equality (49) holds for $t$, and substituting the above equality shows that it also holds at time $t + 1$.

Since the equivalence (49) holds for all $t$, we can take the limit as $t \to \infty$, and doing so yields the claim.

## C.3 Proof of Theorem 3

In the statement of the theorem, we require choosing $\epsilon > 0$ to satisfy the upper bound $\epsilon^2 \precsim \frac{\rho(\epsilon, \delta)}{n}$, and then provide an upper bound in terms of $\sqrt{\rho(\epsilon, \delta)/n}$. It is equivalent to instead choose $\epsilon$ to satisfy the lower bound $\epsilon^2 \succsim \frac{\rho(\epsilon, \delta)}{n}$, and then provide upper bounds proportional to $\epsilon$. For the purposes of the proof, the latter formulation turns out to be more convenient and we pursue it here.

To streamline notation, let us introduce the shorthand $\langle f, \mathcal{D}^\pi(Q) \rangle \overset{def}{=} \langle f, \delta^\pi(Q) \rangle_n - \langle f, \mathcal{B}^\pi(Q) \rangle_\mu$. For each pair $(Q, \pi)$, we then define the random variable

$$Z_n(Q, \pi) \overset{def}{=} \sup_{f \in \mathcal{F}^\pi} \frac{\left| \langle f, \mathcal{D}^\pi(Q) \rangle \right|}{\sqrt{\|f\|_n^2 + \lambda}}.$$

Central to our proof of the theorem is a uniform bound on this random variable, one that holds for all pairs $(Q, \pi)$. In particular, our strategy is to exhibit some $\epsilon > 0$ for which, upon setting $\lambda = 4\epsilon^2$, we have the guarantees

$$\frac{1}{4} \leq \frac{\sqrt{\|f\|_n^2 + \lambda}}{\sqrt{\|f\|_\mu^2 + \lambda}} \leq 2 \qquad \text{uniformly for all } f \in \mathcal{F}, \text{ and} \tag{51a}$$

$$Z_n(Q, \pi) \leq \epsilon \quad \text{uniformly for all } (Q, \pi), \tag{51b}$$

both with probability at least $1 - \delta$. In particular, consistent with the theorem statement, we show that this claim holds if we choose $\epsilon > 0$ to satisfy the inequality

$$\epsilon^2 \geq \bar{c} \frac{\rho(\epsilon, \delta)}{n} \tag{52}$$

where $\bar{c} > 0$ is a sufficiently large (but universal) constant.

Supposing that the bounds (51a) and (51b) hold, let us now establish the set inclusions claimed in the theorem.

**Inclusion** $\mathcal{C}_\infty^\pi \subseteq \widehat{\mathcal{C}}_n^\pi(\epsilon)$**:** Define the random variable $M_n(Q, \pi) \stackrel{def}{=} \sup_{f \in \mathcal{F}^\pi} \frac{|\langle f, \mathcal{B}^\pi(Q) \rangle_\mu|}{\sqrt{\|f\|_\mu^2 + \lambda}}$, and observe that $Q \in \mathcal{C}_\infty^\pi$ implies that $M_n(Q, \pi) = 0$. With this definition, we have

$$\sup_{f \in \mathcal{F}^\pi} \frac{|\langle f, \delta^\pi(Q) \rangle_n|}{\sqrt{\|f\|_n^2 + \lambda}} \overset{(i)}{\leq} M_n(Q, \pi) + Z_n(Q, \pi) \overset{(ii)}{\leq} \epsilon$$

where step (i) follows from the triangle inequality; and step (ii) follows since $M_n(Q, \pi) = 0$, and $Z_n(Q, \pi) \leq \epsilon$ from the bound (51b).

**Inclusion** $\widehat{\mathcal{C}}_n^\pi(\epsilon) \subseteq \mathcal{C}_n^\pi(4\epsilon)$ By the definition of $\mathcal{C}_n^\pi(4\epsilon)$, we need to show that

$$\bar{M}(Q, \pi) \stackrel{def}{=} \sup_{f \in \mathcal{F}^\pi} \frac{|\langle f, \mathcal{B}^\pi(Q) \rangle_\mu|}{\sqrt{\|f\|_\mu^2 + \lambda}} \leq 4\epsilon \qquad \text{for any } Q \in \widehat{\mathcal{C}}_n^\pi(\epsilon).$$

Now we have

$$\bar{M}(Q, \pi) \overset{(i)}{\leq} 2M_n(Q, \pi) \overset{(ii)}{\leq} 2\left\{ \sup_{f \in \mathcal{F}^\pi} \frac{|\langle f, \delta^\pi(Q) \rangle_n|}{\sqrt{\|f\|_n^2 + \lambda}} + Z_n(Q, \pi) \right\} \overset{(iii)}{\leq} 2\{\epsilon + \epsilon\} = 4\epsilon,$$

where step (i) follows from the sandwich relation (51a); step (ii) follows from the triangle inequality and the definition of $Z_n(Q, \pi)$; and step (iii) follows since $Z_n(Q, \pi) \leq \epsilon$ from the bound (51b), and

$$\sup_{f \in \mathcal{F}^\pi} \frac{|\langle f, \delta^\pi(Q) \rangle_n|}{\sqrt{\|f\|_n^2 + \lambda}} \leq \epsilon, \qquad \text{using the inclusion } Q \in \widehat{\mathcal{C}}_n^\pi(\epsilon).$$

Consequently, the remainder of our proof is devoted to establishing the claims (51a) and (51b). In doing so, we make repeated use of some Bernstein bounds, stated in terms of the shorthand $\Psi_n(\delta) = \frac{\log(n/\delta)}{n}$.

**Lemma 10.** *There is a universal constant $c$ such each the following statements holds with probability at least $1 - \delta$. For any $f$, we have*

$$\left| \|f\|_n^2 - \|f\|_\mu^2 \right| \leq c \left\{ \|f\|_\mu \sqrt{\Psi_n(\delta)} + \Psi_n(\delta) \right\}, \tag{53a}$$

*and for any $(Q, \pi)$ and any function $f$, we have*

$$\left| \langle f, \delta^\pi(Q) \rangle_n - \langle f, \mathcal{B}^\pi(Q) \rangle_\mu \right| \leq c \left\{ \|f\|_\mu \sqrt{\Psi_n(\delta)} + \|f\|_\infty \Psi_n(\delta) \right\}. \tag{53b}$$

These bounds follow by identifying a martingale difference sequence, and applying a form of Bernstein's inequality tailored to the martingale setting. See Section C.6.3 for the details.

## C.4 Proof of the sandwich relation (51a)

We claim that (modulo the choice of constants) it suffices to show that

$$\left| \|f\|_n - \|f\|_\mu \right| \leq \epsilon \qquad \text{uniformly for all } f \in \mathcal{F} \tag{54}$$

for some universal constant $c'$. Indeed, when this bound holds, we have

$$\|f\|_n + 2\epsilon \leq \|f\|_\mu + 3\epsilon \leq \frac{3}{2}\{\|f\|_\mu + 2\epsilon\}, \quad \text{and} \quad \|f\|_n + 2\epsilon \geq \|f\|_\mu + \epsilon \geq \frac{1}{2}\{\|f\|_\mu + 2\epsilon\},$$

so that $\frac{\|f\|_\mu + 2\epsilon}{\|f\|_n + 2\epsilon} \in \left[\frac{1}{2}, \frac{3}{2}\right]$. To relate this statement to the claimed sandwich, observe the inclusion $\frac{\|f\| + \sqrt{2}\epsilon}{\sqrt{\|f\|^2 + 4\epsilon^2}} \in [1, \sqrt{2}]$, where $\|f\|$ can be either $\|f\|_n$ or $\|f\|_\mu$. Combining this fact with our previous bound, we see that $\frac{\sqrt{\|f\|_n^2 + 4\epsilon^2}}{\sqrt{\|f\|_\mu^2 + 4\epsilon^2}} \in \left[\frac{1}{\sqrt{2}} \frac{1}{2}, \frac{3\sqrt{2}}{2}\right] \subset \left[\frac{1}{4}, 3\right]$, as claimed.

The remainder of our analysis is focused on proving the bound (54). Defining the random variable $Y_n(f) = \left| \|f\|_n - \|f\|_\mu \right|$, we need to establish a high probability bound on $\sup_{f \in \mathcal{F}} Y_n(f)$. Let

$\{f^1, \ldots, f^N\}$ be an $\epsilon$-cover of $\mathcal{F}$ in the sup-norm. For any $f \in \mathcal{F}$, we can find some $f^j$ such that $\|f - f^j\|_\infty \leq \epsilon$, whence

$$
Y_n(f) \leq Y_n(f^j) + \left|Y_n(f^j) - Y_n(f)\right| \overset{(i)}{\leq} Y_n(f^j) + \left|\|f^j\|_n - \|f\|_n\right| + \left|\|f^j\|_\mu - \|f\|_\mu\right|
$$
$$
\overset{(ii)}{\leq} Y_n(f^j) + \|f^j - f\|_n + \|f^j - f\|_\mu
$$
$$
\overset{(iii)}{\leq} Y_n(f^j) + 2\epsilon,
$$

where steps (i) and (ii) follow from the triangle inequality; and step (iii) follows from the inequality $\max\{\|f^j - f\|_n, \|f^j - f\|_\mu\} \leq \|f^j - f\|_\infty \leq \epsilon$. Thus, we have reduced the problem to bounding a finite maximum.

Note that if $\max\{\|f^j\|_n, \|f^j\|_\mu\} \leq \epsilon$, then we have $Y_n(f^j) \leq 2\epsilon$ by the triangle inequality. Otherwise, we may assume that $\|f^j\|_n + \|f^j\|_n \geq \epsilon$. With probability at least $1 - \delta$, we have

$$
\left|\|f^j\|_n - \|f\|_\mu\right| = \frac{\left|\|f^j\|_n^2 - \|f\|_\mu^2\right|}{\|f^j\|_n + \|f^j\|_\mu} \overset{(i)}{\leq} \frac{c\{\|f^j\|_\mu \sqrt{\Psi_n(\delta)} + \Psi_n(\delta)\}}{\|f^j\|_\mu + \|f^j\|_n}
$$
$$
\overset{(ii)}{\leq} c\left\{\sqrt{\Psi_n(\delta)} + \frac{\Psi_n(\delta)}{\epsilon}\right\},
$$

where step (i) follows from the Bernstein bound (53a) from Lemma 10, and step (ii) uses the fact that $\|f^j\|_n + \|f^j\|_n \geq \epsilon$.

Taking union bound over all $N$ elements in the cover and replacing $\delta$ with $\delta/N$, we have

$$
\max_{j \in [N]} Y_n(f^j) \leq c\left\{\sqrt{\Psi_n(\delta/N)} + \frac{\Psi_n(\delta/N)}{\epsilon}\right\}
$$

with probability at least $1 - \delta$. Recalling that $N = N_\epsilon(\mathcal{F})$, our choice (52) of $\epsilon$ ensures that $\sqrt{\Psi_n(\delta/N)} \leq c\,\epsilon$ for some universal constant $c$. Putting together the pieces (and increasing the constant $\bar{c}$ in the choice (52) of $\epsilon$ as needed) yields the claim.

## C.5 Proof of the uniform upper bound (51b)

We need to establish an upper bound on $Z_n(Q, \pi)$ that that holds uniformly for all $(Q, \pi)$. Our first step is to prove a high probability bound for a fixed pair. We then apply a standard discretization argument to make it uniform in the pair.

Note that we can write $Z_n(Q, \pi) = \sup_{f \in \mathcal{F}} \frac{V_n(f)}{\sqrt{\|f\|_n^2 + \lambda}}$, where we have defined $V_n(f) \overset{def}{=} |\langle f, \mathcal{D}^\pi(Q)\rangle|$. Our first lemma provides a uniform bound on the latter random variables:

**Lemma 11.** *Suppose that* $\epsilon^2 \geq \Psi_n\big(\delta/N_\epsilon(\mathcal{F})\big)$. *Then we have*

$$
V_n(f) \leq c\big\{\|f\|_\mu \epsilon + \epsilon^2\big\} \qquad \text{for all } f \in \mathcal{F} \tag{55}
$$

*with probability at least* $1 - \delta$.

See Appendix C.6.1 for the proof of this claim.

We claim that the bound (55) implies that, for any fixed pair $(Q, \pi)$, we have

$$
Y_n(Q, \pi) \leq c'\epsilon \qquad \text{with probability at least } 1 - \delta.
$$

Indeed, when Lemma 11 holds, for any $f \in \mathcal{F}$, we can write

$$
\frac{V_n(f)}{\sqrt{\|f\|_n^2 + \lambda}} = \frac{\sqrt{\|f\|_\mu^2 + \lambda}}{\sqrt{\|f\|_n^2 + \lambda}} \frac{V_n(f)}{\sqrt{\|f\|_\mu^2 + \lambda}} \overset{(i)}{\leq} 3 \frac{c\{\|f\|_\mu \epsilon + \epsilon^2\}}{\sqrt{\|f\|_\mu^2 + \lambda}} \overset{(ii)}{\leq} c'\epsilon,
$$

where step (i) uses the sandwich relation (51a), along with the bound (55); and step (ii) follows given the choice $\lambda = 4\epsilon^2$. We have thus proved that for any fixed $(Q, \pi)$ and $\epsilon \geq \Psi_n\big(\delta/N_\epsilon(\mathcal{F})\big)$, we have

$$
Z_n(Q, \pi) \leq c'\epsilon \qquad \text{with probability at least } 1 - \delta. \tag{56}
$$

1176 Our next step is to upgrade this bound to one that is uniform over all pairs $(Q, \pi)$. We do so via a
1177 discretization argument: let $\{Q^j\}_{j=1}^J$ and $\{\pi^k\}_{k=1}^K$ be $\epsilon$-coverings of $\mathcal{Q}$ and $\Pi$, respectively.

1178 **Lemma 12.** *We have the upper bound*

$$\sup_{Q,\pi} Z_n(Q, \pi) \leq \max_{(j,k) \in [J] \times [K]} Z_n(Q^j, \pi^k) + 4\epsilon. \tag{57}$$

1179 See Section C.6.2 for the proof of this claim.

1180 If we replace $\delta$ with $\delta/(JK)$, then we are guaranteed that the bound (56) holds uniformly over the
1181 family $\{Q^j\}_{j=1}^J \times \{\pi^k\}_{k=1}^K$. Recalling that $J = N_\epsilon(\mathcal{Q})$ and $K = N_\epsilon(\Pi)$, we conclude that for any
1182 $\epsilon$ satisfying the inequality (52), we have $\sup_{Q,\pi} Z_n(Q, \pi) \leq \tilde{c}\epsilon$ with probability at least $1 - \delta$. (Note
1183 that by suitably scaling up $\epsilon$ via the choice of constant $\bar{c}$ in the bound (52), we can arrange for $\tilde{c} = 1$,
1184 as in the stated claim.)

## C.6 Proofs of supporting lemmas

1186 In this section, we collect together the proofs of Lemmas 11 and 12, which were stated and used
1187 in Appendix C.5.

### C.6.1 Proof of Lemma 11

1189 We first localize the problem to the class $\mathcal{F}(\epsilon) = \{f \in \mathcal{F} \mid \|f\|_\mu \leq \epsilon\}$. In particular, if there exists
1190 some $\tilde{f} \in \mathcal{F}$ that violates (55), then the rescaled function $f = \epsilon \tilde{f}/\|\tilde{f}\|_\mu$ belongs to $\mathcal{F}(\epsilon)$, and satisfies
1191 $V_n(f) \geq c\epsilon^2$. Consequently, it suffices to show that $V_n(f) \leq c\epsilon^2$ for all $f \in \mathcal{F}(\epsilon)$.

1192 Choose an $\epsilon$-cover of $\mathcal{F}$ in the sup-norm with $N = N_\epsilon(\mathcal{F})$ elements. Using this cover, for any
1193 $f \in \mathcal{F}(\epsilon)$, we can find some $f^j$ such that $\|f - f^j\|_\infty \leq \epsilon$. Thus, for any $f \in \mathcal{F}(\epsilon)$, we can write

$$V_n(f) \leq V_n(f^j) + V_n(f - f^j) \leq \underbrace{V_n(f^j)}_{T_1} + \underbrace{\sup_{g \in \mathcal{G}(\epsilon)} V_n(g)}_{T_2}, \tag{58}$$

1194 where $\mathcal{G}(\epsilon) \stackrel{def}{=} \{f_1 - f_2 \mid f_1, f_2 \in \mathcal{F}, \|f_1 - f_2\|_\infty \leq \epsilon\}$. We bound each of these two terms in turn.
1195 In particular, we show that each of $T_1$ and $T_2$ are upper bounded by $c\epsilon^2$ with high probability.

**Bounding $T_1$:** From the Bernstein bound (53b), we have

$$V_n(f^k) \leq c\{\|f^k\|_\mu \sqrt{\Psi_n(\delta/N)} + \|f^k\|_\infty \Psi_n(\delta/N)\} \qquad \text{for all } k \in [N]$$

1197 with probability at least $1 - \delta$. Now for the particular $f^j$ chosen to approximate $f \in \mathcal{F}(\epsilon)$, we have

$$\|f^j\|_\mu \leq \|f^j - f\|_\mu + \|f\|_\mu \leq 2\epsilon,$$

1198 where the inequality follows since $\|f^j - f\|_\mu \leq \|f^j - f\|_\infty \leq \epsilon$, and $\|f\|_\mu \leq \epsilon$. Consequently, we
1199 conclude that

$$T_1 \leq c\Big\{2\epsilon\sqrt{\Psi_n(\delta/N)} + \Psi_n(\delta/N)\Big\} \leq c'\epsilon^2 \qquad \text{with probability at least } 1 - \delta.$$

1200 where the final inequality follows from our choice of $\epsilon$.

**Bounding $T_2$:** Define $\mathcal{G} \stackrel{def}{=} \{f_1 - f_2 \mid f_1, f_2 \in \mathcal{F}\}$. We need to bound a supremum of the
1202 process $\{V_n(g), g \in \mathcal{G}\}$ over the subset $\mathcal{G}(\epsilon)$. From the Bernstein bound (53b), the increments
1203 $V_n(g_1) - V_n(g_2)$ of this process are sub-Gaussian with parameter $\|g_1 - g_2\|_\mu \leq \|g_1 - g_2\|_\infty$, and
1204 sub-exponential with parameter $\|g_1 - g_2\|_\infty$. Therefore, we can apply a chaining argument that uses
1205 the metric entropy $\log N_t(\mathcal{G})$ in the supremum norm. Moreover, we can terminate the chaining at $2\epsilon$,
1206 because we are taking the supremum over the subset $\mathcal{G}(\epsilon)$, and it has sup-norm diameter at most $2\epsilon$.
1207 Moreover, the lower interval of the chain can terminate at $2\epsilon^2$, since our goal is to prove an upper
1208 bound of this order. Then, by using high probability bounds for the suprema of empirical processes
1209 (e.g., Theorem 5.36 in the book [Wai19]), we have

$$T_2 \leq c_1 \int_{2\epsilon^2}^{2\epsilon} \phi\Big(\frac{\log N_t(\mathcal{G})}{n}\Big) dt + c_2\{\epsilon\sqrt{\Psi_n(\delta)} + \epsilon\Psi_n(\delta)\} + 2\epsilon^2$$

1210 with probability at least $1 - \delta$. (Here the reader should recall our shorthand $\phi(s) = \max\{s, \sqrt{s}\}$.)

1211 Since $\mathcal{G}$ consists of differences from $\mathcal{F}$, we have the upper bound $\log N_t(\mathcal{G}) \leq 2 \log N_{t/2}(\mathcal{F})$, and
1212 hence (after making the change of variable $u = t/2$ in the integrals)

$$T_2 \leq c_1' \int_{\epsilon^2}^{\epsilon} \phi\left(\frac{\log N_u(\mathcal{F})}{n}\right) du + c_2\left\{\epsilon\sqrt{\Psi_n(\delta)} + \epsilon\Psi_n(\delta)\right\} \leq \tilde{c}\epsilon^2,$$

1213 where the last inequality follows from our choice of $\epsilon$.

### C.6.2  Proof of Lemma 12

1215 By our choice of the $\epsilon$-covers, for any $(Q, \pi)$, there is a pair $(Q^j, \pi^k)$ such that

$$\|Q^j - Q\|_\infty \leq \epsilon, \quad \text{and} \quad \|\pi^k - \pi\|_{\infty,1} = \sup_s \|\pi^k(\cdot \mid s) - \pi(\cdot \mid s)\|_1 \leq \epsilon.$$

1216 Using this pair, an application of the triangle inequality yields

$$\left|Z_n(Q,\pi) - Z_n(Q^j,\pi^k)\right| \leq \underbrace{\left|Z_n(Q,\pi) - Z_n(Q,\pi^k)\right|}_{T_1} + \underbrace{\left|Z_n(Q,\pi^k) - Z_n(Q^j,\pi^k)\right|}_{T_2}$$

1217 We bound each of these terms in turn, in particular proving that $T_1 + T_2 \leq 24\epsilon$. Putting together the
1218 pieces yields the bound stated in the lemma.

**Bounding $T_2$:**  From the definition of $Z_n$, we have

$$T_2 = \left|Z_n(Q,\pi^k) - Z_n(Q^j,\pi^k)\right| \leq \sup_{f \in \mathcal{F}} \frac{|\langle f, \mathcal{D}^{\pi^k}(Q - Q^j)\rangle|}{\sqrt{\|f\|_n^2 + \lambda}}.$$

1220 Now another application of the triangle inequality yields

$$\begin{aligned}
|\langle f, \mathcal{D}^{\pi^k}(Q - Q^j)\rangle| &\leq |\langle f, \delta^{\pi^k}(Q - Q^j)\rangle_n| + |\langle f, \mathcal{B}^{\pi^k}(Q - Q^j)\rangle|_\mu \\
&\leq \|f\|_n \|\delta^{\pi^k}(Q - Q^j)\|_n + \|f\|_\mu \|\mathcal{B}^{\pi^k}(Q - Q^j)\|_\mu \\
&\leq \max\{\|f\|_n, \|f\|_\mu\} \left\{\|\delta^{\pi^k}(Q - Q^j)\|_\infty + \|\mathcal{B}^{\pi^k}(Q - Q^j)\|_\infty\right\}
\end{aligned}$$

1221 where step (i) follows from the Cauchy–Schwarz inequality.  Now in terms of the shorthand
1222 $\Delta \overset{def}{=} Q - Q^j$, we have

$$\|\mathcal{B}^{\pi^k}(Q - Q^j)\|_\infty = \sup_{(s,a)} \left|\Delta(s,a) - \gamma\mathbb{E}_{s^+\sim\mathbb{P}(s,a)}\left[\Delta(s^+,\pi)\right]\right| \leq 2\|\Delta\|_\infty \leq 2\epsilon. \tag{59a}$$

1223 An entirely analogous argument yields

$$\|\delta^{\pi^k}(Q - Q^j)\|_\infty \leq 2\epsilon \tag{59b}$$

1224 Conditioned on the sandwich relation (51a), we have $\sup_{f\in\mathcal{F}} \frac{\max\{\|f\|_n, \|f\|_\mu\}}{\sqrt{\|f\|_n^2 + \lambda}} \leq 4$. Combining this
1225 bound with inequalities (59a) and (59b), we have shown that $T_2 \leq 4\{2\epsilon + 2\epsilon\} = 16\epsilon$.

**Bounding $T_1$:**  In this case, a similar argument yields

$$|\langle f, (\mathcal{D}^\pi - \mathcal{D}^{\pi^k})(Q)\rangle| \leq \max\{\|f\|_n, \|f\|_\mu\} \left\{\|(\delta^\pi - \delta^{\pi^k})(Q)\|_n + \|(\mathcal{B}^\pi - \mathcal{B}^{\pi^k})(Q)\|_\mu\right\}.$$

1227 Now we have

$$\begin{aligned}
\|(\delta^\pi - \delta^{\pi^k})(Q)\|_n &\leq \max_{i=1,\ldots,n} \left|\sum_{a'} \left(\pi(a' \mid s_i) - \pi^k(a' \mid s_i)\right)Q(s_i^+, a')\right| \\
&\leq \max_s \sum_{a'} |\pi(a' \mid s) - \pi^k(a \mid s)| \, \|Q\|_\infty \\
&\leq \epsilon.
\end{aligned}$$

1228 A similar argument yields that $\|(\mathcal{B}^\pi - \mathcal{B}^{\pi^k})(Q)\|_\mu| \leq \epsilon$, and arguing as before, we conclude that
1229 $T_1 \leq 4\{\epsilon + \epsilon\} = 8\epsilon$.

### C.6.3 Proof of Lemma 10

Our proof of this claim makes use of the following known Bernstein bound for martingale differences (cf. Theorem 1 in the paper [BLL$^+$11]). Recall the shorthand notation $\Psi_n(\delta) = \frac{\log(n/\delta)}{n}$.

**Lemma 13** (Bernstein's Inequality for Martingales). *Let $\{X_t\}_{t\geq 1}$ be a martingale difference sequence with respect to the filtration $\{\mathcal{F}_t\}_{t\geq 1}$. Suppose that $|X_t| \leq 1$ almost surely, and let $\mathbb{E}_t$ denote expectation conditional on $\mathcal{F}_t$. Then for all $\delta \in (0,1)$, we have*

$$\Big|\frac{1}{n}\sum_{t=1}^{n} X_t\Big| \leq 2\Big[\Big(\frac{1}{n}\sum_{t=1}^{n}\mathbb{E}_t X_t^2\Big)\Psi_n(2\delta)\Big]^{1/2} + 2\Psi_n(2\delta) \tag{60}$$

*with probability at least $1 - \delta$.*

With this result in place, we divide our proof into two parts, corresponding to the two claims (53b) and (53a) stated in Lemma 10.

**Proof of the bound** (53b): Recall that at step $i$, the triple $(s, a, o)$ is drawn according to a conditional distribution $\mu_i(\cdot \mid \mathcal{F}_i)$. Similarly, we let $d_i$ denote the distribution of $(s, a, r, s^+, o)$ conditioned on the filtration $\mathcal{F}_i$. Note that $\mu_i$ is obtained from $d_i$ by marginalizing out the pair $(r, s^+)$. Moreover, by the tower property of expectation, the Bellman error is equivalent to the average TD error.

Using these facts, we have the equivalence

$$\begin{aligned}
\langle f, \delta^\pi Q\rangle_{d_i} &= \mathbb{E}_{d_i}\big\{f(s,a,o)[Q(s,a) - r - \gamma Q(s^+,\pi)]\big\}\\
&= \mathbb{E}_{(s,a,o)\sim\mu_i}\big\{f(s,a,o)\mathbb{E}_{r\sim R(s,a),s^+\sim\mathbb{P}(s,a)}[Q(s,a) - r - \gamma Q(s^+,\pi)]\big\}\\
&= \mathbb{E}_{(s,a,o)\sim\mu_i}\big\{f(s,a,o)[Q(s,a) - (\mathcal{T}^\pi Q)(s,a)]\big\}\\
&= \langle f, \mathcal{B}^\pi Q\rangle_{\mu_i}.
\end{aligned}$$

As a consequence, we can write $\langle f, \delta^\pi(Q)\rangle_n - \langle f, \mathcal{B}^\pi(Q)\rangle_\mu = \frac{1}{n}\sum_{i=1}^{n} W_i$ where

$$W_i \overset{def}{=} f(s_i,a_i,o_i)[Q(s_i,a_i) - r_i - \gamma Q(s_i^+,\pi)] - \mathbb{E}_{d_i}\big\{f(s,a,o)[Q(s,a) - r - \gamma Q(s^+,\pi)]\big\}$$

defines a martingale difference sequence (MDS). Thus, we can prove the claim by applying a Bernstein martingale inequality.

Since $\|r\|_\infty \leq 1$ and $\|Q\|_\infty \leq 1$ by assumption, we have $\|W_i\|_\infty \leq 3\|f\|_\infty$, and

$$\frac{1}{n}\sum_{i=1}^{n}\mathbb{E}_{d_i}[W_i^2] \leq 9\,\frac{1}{n}\sum_{i=1}^{n}\mathbb{E}_{\mu_i}[f^2(s_i,a_i,o_i)] \;=\; 9\|f\|_\mu^2.$$

Consequently, the claimed bound (53b) follows by applying the Bernstein bound stated in Lemma 13.

**Proof of the bound** (53a): In this case, we have the additive decomposition

$$\|f\|_n^2 - \|f\|_\mu^2 = \frac{1}{n}\sum_{i=1}^{n}\Big\{\underbrace{f^2(s_i,a_i,o_i) - \mathbb{E}_{\mu_i}[f^2(s,a,o)]}_{W_i'}\Big\},$$

where $\{W_i'\}_{i=1}^{n}$ again defines a martingale difference sequence. Note that $\|W_i'\|_\infty \leq 2\|f\|_\infty^2 \leq 2$, and

$$\frac{1}{n}\sum_{i=1}^{n}\mathbb{E}_{\mu_i}[(W_i')^2] \overset{(i)}{\leq} \frac{1}{n}\sum_{i=1}^{n}\mathbb{E}_{\mu_i}\big[f^4(S,A,O)\big] \;\leq\; \|f\|_\infty^2\frac{1}{n}\sum_{i=1}^{n}\mathbb{E}_{\mu_i}\big[f^2(S,A,O)\big] \overset{(ii)}{\leq} \|f\|_\mu^2,$$

where step (i) uses the fact that the variance of $f^2$ is at most the fourth moment, and step (ii) uses the bound $\|f\|_\infty \leq 1$. Consequently, the claimed bound (53a) follows by applying the Bernstein bound stated in Lemma 13.

# D    Proofs for Section 4 and Appendix A.5

In this section, we collect together the proofs of results stated without proof in Section 4 and Appendix A.5.

## D.1    Proof of Proposition 1

*Proof.* Since $f^* \in \mathcal{F}^\pi$, we are guaranteed that the corresponding constraint must hold. It reads as

$$|\mathbb{E}_\mu \frac{1}{b_\pi} \frac{d_\pi}{\mu} \mathcal{B}^\pi Q|^2 = \frac{1}{b_\pi^2} |\mathbb{E}_\pi \mathcal{B}^\pi Q|^2 \overset{(iii)}{\leq} \left( \frac{1}{b_\pi^2} \|\frac{d_\pi}{\mu}\|_\mu^2 + \lambda \right) \frac{\rho}{n}.$$

where step (iii) follows from the definition of population constraint. Re-arranging yields the upper bound

$$\frac{|\mathbb{E}_\mu \frac{d_\pi}{\mu} \mathcal{B}^\pi Q|^2}{(1+\lambda)\frac{\rho}{n}} \leq \frac{\left( \|\frac{d_\pi}{\mu}\|_\mu^2 + b_\pi^2 \lambda \right) \frac{\rho}{n}}{(1+\lambda)\frac{\rho}{n}} = \frac{\mathbb{E}_\pi \left[ \frac{d_\pi(S,A)}{\mu(S,A)} \right] + b_\pi^2 \lambda}{1+\lambda},$$

where the final step uses the fact that

$$\|\frac{d_\pi}{\mu}\|_\mu^2 = \mathbb{E}_\mu \frac{d_\pi^2(S,A)}{\mu^2(S,A)} = \mathbb{E}_\pi \frac{d_\pi(S,A)}{\mu(S,A)}$$

Thus, we have established the bound (i) in our claim (12).

The upper bound (ii) follows immediately since $\mathbb{E}_\pi \frac{d_\pi(s,a)}{\mu(s,a)} \leq \sup_{(s,a)} \frac{d_\pi(s,a)}{\mu(s,a)} \leq b_\pi$.

$\square$

## D.2    Proof of Lemma 1

Some simple algebra yields

$$\mathcal{B}^\pi Q - \mathcal{B}^\pi Q_\star^\pi = [Q - \mathcal{T}^\pi Q] - [Q_\star^\pi - \mathcal{T}^\pi Q_\star^\pi] = (\mathcal{I} - \gamma \mathbb{P}^\pi)(Q - Q_\star^\pi) = (\mathcal{I} - \gamma \mathbb{P}^\pi)\epsilon.$$

Taking expectations under $\pi$ and recalling that $\langle f, \mathcal{B}^\pi Q_\star^\pi \rangle_\pi = 0$ for all $f \in \mathcal{F}^\pi$ yields

$$\langle f, \mathcal{B}^\pi Q \rangle_\pi = \langle f, (\mathcal{I} - \gamma \mathbb{P}^\pi)\epsilon \rangle_\pi.$$

Notice that for any $Q \in \mathcal{Q}^\pi$ there exists a test function $\epsilon = Q - Q_\star^\pi \in \mathcal{E}^\pi$, and the associated population constraint reads

$$\frac{\left| \langle \epsilon, (\mathcal{I} - \gamma \mathbb{P}^\pi)\epsilon \rangle_\mu \right|}{\sqrt{\|\epsilon\|_\mu^2 + \lambda}} \leq \sqrt{\frac{\rho}{n}}.$$

Consequently, the off-policy cost coefficient can be upper bounded as

$$K^\pi \leq \max_{\epsilon \in \mathcal{E}_\star^\pi} \left\{ \frac{\rho}{n} \frac{\langle \mathbb{1}, (\mathcal{I} - \gamma \mathbb{P}^\pi)\epsilon \rangle_\pi^2}{1+\lambda} \right\} \leq \max_{\epsilon \in \mathcal{E}_\star^\pi} \left\{ \frac{\|\epsilon\|_\mu^2 + \lambda}{\|\mathbb{1}\|_\pi^2 + \lambda} \frac{\langle \mathbb{1}, (\mathcal{I} - \gamma \mathbb{P}^\pi)\epsilon \rangle_\pi^2}{\langle \epsilon, (\mathcal{I} - \gamma \mathbb{P}^\pi)\epsilon \rangle_\mu^2} \right\},$$

as claimed in the bound (14).

## D.3    Proof of Lemma 7

If weak Bellman closure holds, then we can write

$$\mathcal{B}^\pi Q = Q - \mathcal{T}^\pi Q = Q - \mathcal{P}^\pi(Q) \in \mathcal{E}^\pi.$$

For any $Q \in \mathcal{Q}^\pi$, the function $\epsilon = Q - \mathcal{P}^\pi(Q)$ belongs to $\mathcal{E}^\pi$, and the associated population constraint reads $\frac{|\langle \epsilon, \epsilon \rangle_\mu|}{\sqrt{\|\epsilon\|_\mu^2 + \lambda}} \leq \sqrt{\frac{\rho}{n}}$. Consequently, the off-policy cost coefficient is upper bounded as

$$K^\pi \leq \max_{\epsilon \in \mathcal{E}^\pi} \left\{ \frac{n}{\rho} \frac{v \langle \mathbb{1}, \epsilon \rangle_\pi^2}{1+\lambda} \right\} \leq \max_{\epsilon \in \mathcal{E}^\pi} \left\{ \frac{\|\epsilon\|_\mu^2 + \lambda}{1+\lambda} \frac{\langle \mathbb{1}, \epsilon \rangle_\pi^2}{\langle \epsilon, \epsilon \rangle_\mu^2} \right\} \leq \max_{\epsilon \in \mathcal{E}^\pi} \left\{ \frac{\langle \mathbb{1}, \epsilon \rangle_\pi^2}{\langle \epsilon, \epsilon \rangle_\mu^2} \right\},$$

where the final inequality follows from the fact that $\|\epsilon\|_\mu \leq 1$.

### D.4  Proof of Lemma 2

We split our proof into the two separate claims.

**Proof of the bound** (16a): When the test function class includes $\mathcal{F}_\pi^{\mathcal{B}}$, then any $Q$ feasible must satisfy the population constraints

$$\frac{\langle \mathcal{B}^\pi Q', \mathcal{B}^\pi Q \rangle_\mu}{\sqrt{\|\mathcal{B}^\pi Q'\|_\mu^2 + \lambda}} \leq \sqrt{\frac{\rho}{n}}, \qquad \text{for all } Q' \in Q^\pi.$$

Setting $Q' = Q$ yields $\frac{\|\mathcal{B}^\pi Q\|_\mu^2}{\sqrt{\|\mathcal{B}^\pi Q\|_\mu^2 + \lambda}} \leq \sqrt{\frac{\rho}{n}}$. If $\|\mathcal{B}^\pi Q\|_\mu^2 \geq \lambda$, then the claim holds, given our choice $\lambda = c\frac{\rho}{n}$ for some constant $c$. Otherwise, the constraint can be weakened to $\frac{\|\mathcal{B}^\pi Q\|_\mu^2}{\sqrt{2\|\mathcal{B}^\pi Q\|_\mu^2}} \leq \sqrt{\frac{\rho}{n}}$, which yields the bound (16a).

**Proof of the bound** (16b): We now prove the sequence of inequalities stated in equation (16b). Inequality (i) follows directly from the definition of $K^\pi$ and Lemma 2. Turning to inequality (ii), an application of Jensen's inequality yields

$$\langle \mathbb{1}, \mathcal{B}^\pi Q \rangle_\pi^2 = [\mathbb{E}_\pi \mathcal{B}^\pi Q]^2 \leq \mathbb{E}_\pi [\mathcal{B}^\pi Q]^2 = \|\mathcal{B}^\pi Q\|_\pi^2.$$

Finally, inequality (iii) follows by observing that

$$\sup_{Q \in \mathcal{Q}^\pi} \frac{\|\mathcal{B}^\pi Q\|_\pi^2}{\|\mathcal{B}^\pi Q\|_\mu^2} = \sup_{Q \in \mathcal{Q}^\pi} \frac{\mathbb{E}_\pi [(\mathcal{B}^\pi Q)(s,a)]^2}{\mathbb{E}_\mu [(\mathcal{B}^\pi Q)(s,a)]^2} = \sup_{Q \in \mathcal{Q}^\pi} \frac{\mathbb{E}_\mu \left[ \frac{d_\pi(s,a)}{\mu(s,a)} \right] [(\mathcal{B}^\pi Q)(s,a)]^2}{\mathbb{E}_\mu [(\mathcal{B}^\pi Q)(s,a)]^2} \leq \sup_{(s,a)} \frac{d_\pi(s,a)}{\mu(s,a)}.$$

## E  Proofs for the Linear Setting

We now prove the results stated in Section 5. Throughout this section, the reader should recall that $Q$ takes the linear function $Q(s,a) = \langle w, \phi(s,a)\rangle$, so that the bulk of our arguments operate directly on the weight vector $w \in \mathbb{R}^d$.

Given the linear structure, the population and empirical covariance matrices of the feature vectors play a central role. We make use of the following known result (cf. Lemma 1 in the paper [ZJZ21]) that relates these objects:

**Lemma 14** (Covariance Concentration). *There are universal constants $(c_1, c_2, c_3)$ such that for any $\delta \in (0,1)$, we have*

$$c_1 \mathbb{E}_\mu \phi\phi^\top \preceq \frac{1}{n}\sum_{i=1}^n \phi_i \phi_i^\top + \frac{c_2}{n}\log\frac{nd}{\delta} I \preceq c_3 \mathbb{E}_\mu \phi\phi^\top + \frac{c_4}{n}\log\frac{nd}{\delta}I. \tag{61}$$

*with probability at least $1 - \delta$.*

### E.1  Proof of Proposition 2

Under weak realizability, we have

$$\langle f_j, \mathcal{B}^\pi Q_\star^\pi\rangle_\mu = 0 \qquad \text{for all } j = 1,\dots,d. \tag{62}$$

Thus, at $(s,a)$ the Bellman error difference reads

$$\begin{aligned}
\mathcal{B}^\pi Q(s,a) - \mathcal{B}^\pi Q_\star^\pi(s,a) &= [Q - \mathcal{T}^\pi Q](s,a) - [Q_\star^\pi - \mathcal{T}^\pi Q_\star^\pi](s,a) \\
&= [Q - Q_\star^\pi](s,a) - \gamma\mathbb{E}_{s^+ \sim \mathbb{P}(s,a)}[Q - Q_\star^\pi](s^+,\pi) \\
&= \langle w - w_\star^\pi, \phi(s,a) - \gamma\phi^{+\pi}(s,a)\rangle
\end{aligned} \tag{63}$$

To proceed we need the following auxiliary result:

**Lemma 15** (Linear Parameter Constraints). *With probability at least $1 - \delta$, there exists a universal constant $c_1 > 0$ such that if $Q \in \mathcal{C}_n^\pi$ then $\|w - w_\star^\pi\|_{\Sigma_{\lambda,Boot}^{+\pi}}^2 \leq c_1\frac{d\rho}{n}$.*

See Appendix E.2 for the proof.

Using this lemma, we can bound the OPC coefficient as follows

$$\begin{aligned}
K^\pi &\overset{(i)}{\leq} \frac{n}{\rho}\max_{Q \in \mathcal{C}_n^\pi}\langle \mathbb{1}, \mathcal{B}^\pi Q - \mathcal{B}^\pi Q_\star^\pi\rangle_\pi^2 \overset{(ii)}{\leq} \frac{n}{\rho}\left[\mathbb{E}_\pi(\phi - \gamma\phi^{+\pi})^\top(w - w_\star^\pi)\right]^2 \\
&\overset{(iii)}{\leq} \frac{n}{\rho}\|\mathbb{E}_\pi\phi - \gamma\phi^{+\pi}\|_{(\Sigma_{\lambda,Boot}^{+\pi})^{-1}}^2\|w - w_\star^\pi\|_{\Sigma_{\lambda,Boot}^{+\pi}}^2 \\
&\leq c_1 d\|\mathbb{E}_\pi\phi - \gamma\phi^{+\pi}\|_{(\Sigma_{\lambda,Boot}^{+\pi})^{-1}}^2.
\end{aligned}$$

Here step $(i)$ follows from the definition of off-policy cost coefficient, $(ii)$ leverages the linear structure and $(iii)$ is Cauchy-Schwartz.

### E.2  Proof of Lemma 15

Under the event of Theorem 3, the statement of Eq. (51a) holds, and in particular

$$\frac{1}{c_1(\sqrt{\|f\|_\mu^2 + \lambda})} \geq \frac{1}{\sqrt{\|f\|_n^2 + \lambda}} \geq \frac{1}{c_2(\sqrt{\|f\|_\mu^2 + \lambda})}.$$

Thus, the $j$ constraint reads

$$\frac{L}{\sqrt{n}} \gtrsim \frac{\langle f_j, \mathcal{B}^\pi Q\rangle_\mu}{\sqrt{\|f\|_n^2 + \lambda}} = \frac{\langle f_j, \mathcal{B}^\pi Q\rangle_\mu}{\sqrt{\widehat{\lambda}_j + \lambda}}$$

where the last step follows from

$$\|f_j\|_{\mathcal{D}}^2 = \frac{1}{n} \sum_{(s,a,r,s^+) \in \mathcal{D}} (f_j(s,a))^2 = \frac{1}{n} \sum_{i=1}^{n} (\widehat{u}_j^\top \phi_i)^2 = \widehat{u}_j^\top \widehat{\Sigma} \widehat{u}_j = \widehat{\lambda}_j.$$

Now, squaring and summing over the constraints and using Eq. (63) yields

$$\begin{aligned}
d\frac{L^2}{n} &\gtrsim \sum_{j=1}^{m} \langle \frac{\widehat{u}_j^\top \phi}{\sqrt{\widehat{\lambda}_j + \lambda}}, (\phi - \gamma\phi^{+\pi})^\top (w - w_\star^\pi) \rangle_\mu^2 \\
&= \sum_{j=1}^{m} \Big[ \frac{\widehat{u}_j^\top}{\sqrt{\widehat{\lambda}_j + \lambda}} \mathbb{E}_\mu \phi(\phi - \gamma\phi^{+\pi})^\top (w - w_\star^\pi) \Big]^2 \\
&= \sum_{j=1}^{m} \Big[ \frac{\widehat{u}_j^\top}{\sqrt{\widehat{\lambda}_j + \lambda}} \underbrace{(\Sigma - \gamma\Sigma^{+\pi})(w - w_\star^\pi)}_{\overset{def}{=} y} \Big]^2 \\
&= y^\top \Big( \sum_{j=1}^{m} \frac{\widehat{u}_j \widehat{u}_j^\top}{\widehat{\lambda}_j + \lambda} \Big) y \\
&= y^\top \Big( \widehat{\Sigma} + \lambda I \Big)^{-1} y \\
&\gtrsim y^\top \Sigma_\lambda^{-1} y.
\end{aligned}$$

The last inequality holds via Lemma 14 *(Covariance Concentration)* with probability at least $1 - \delta$ since $\lambda$ is a large enough regularizer. Let us complete the quadratic form:

$$\|y + \lambda(w - w_\star^\pi)\|_{\Sigma_\lambda^{-1}}^2 \le (\|y\|_{\Sigma_\lambda^{-1}} + \lambda\|(w - w_\star^\pi)\|_{\Sigma_\lambda^{-1}})^2 \lesssim \|y\|_{\Sigma_\lambda^{-1}}^2 + \lambda.$$

Therefore, adding $\lambda$ to both sides of the prior display and noticing that $\lambda \lesssim \frac{L^2}{n}$ gives

$$\begin{aligned}
d\frac{L^2}{n} &\gtrsim \|y + \lambda(w - w_\star^\pi)\|_{\Sigma_\lambda^{-1}}^2 \\
&= (w - w_\star^\pi)(\Sigma_\lambda - \gamma\Sigma^{+\pi})^\top \Big( \Sigma_\lambda^{-1} \Big)(\Sigma_\lambda - \gamma\Sigma^{+\pi})(w - w_\star^\pi) \\
&= (w - w_\star^\pi)(\Sigma_{\lambda,\text{Boot}}^{+\pi})(w - w_\star^\pi) \\
&= \|(w - w_\star^\pi)\|_{\Sigma_{\lambda,\text{Boot}}^{+\pi}}^2.
\end{aligned}$$

## E.3 Proof of Proposition 3

Under weak Bellman closure, we have

$$\mathcal{B}^\pi Q = Q - \mathcal{T}^\pi Q = \phi^\top(w - \mathcal{P}^\pi(w)). \tag{64}$$

With a slight abuse of notation, let $\mathcal{P}^\pi(w)$ denote the weight vector that defines the action-value function $\mathcal{P}^\pi(Q)$. We introduce the following auxiliary lemma:

**Lemma 16** (Linear Parameter Constraints with Bellman Closure)**.** *With probability at least $1 - \delta$, if $Q \in \mathcal{C}_n^\pi$ then $\|w - \mathcal{P}^\pi(w)\|_{\Sigma_\lambda}^2 \le c_1 \frac{d\rho}{n}$.*

See Appendix E.4 for the proof. Using this lemma, we can bound the OPC coefficient as follows

$$\begin{aligned}
K^\pi &\overset{(i)}{\le} \frac{n}{\rho} \max_{Q \in \mathcal{C}_n^\pi} \langle \mathbb{1}, \mathcal{B}^\pi Q \rangle_\pi^2 \overset{(ii)}{\le} \frac{n}{\rho} [\mathbb{E}_\pi(\phi)^\top (w - \mathcal{P}^\pi(w))]^2 \\
&\overset{(iii)}{\le} \frac{n}{\rho} \|\mathbb{E}_\pi \phi\|_{(\Sigma_\lambda)^{-1}}^2 \|w - \mathcal{P}^\pi(w)\|_{\Sigma_\lambda}^2 \\
&\le c_1 d \|\mathbb{E}_\pi \phi\|_{(\Sigma_\lambda)^{-1}}^2.
\end{aligned}$$

Here step $(i)$ follows from the definition of off-policy cost coefficient, $(ii)$ leverages the linear structure and $(iii)$ is Cauchy-Schwartz.

 ## E.4 Proof of Appendix E.4

 Under the event of Theorem 3, the statement of Eq. (51a) holds, and in particular

$$\frac{1}{c_1(\sqrt{\|f\|_\mu^2 + \lambda})} \geq \frac{1}{\sqrt{\|f\|_n^2 + \lambda}} \geq \frac{1}{c_2(\sqrt{\|f\|_\mu^2 + \lambda})}.$$

 Thus, the $j$ constraint reads

$$\frac{L}{\sqrt{n}} \gtrsim \frac{\langle f_j, \mathcal{B}^\pi Q \rangle_\mu}{\sqrt{\|f\|_n^2 + \lambda}} = \frac{\langle f_j, \mathcal{B}^\pi Q \rangle_\mu}{\sqrt{\widehat{\lambda}_j + \lambda}}$$

 where the last step follows from

$$\|f_j\|_{\mathcal{D}}^2 = \frac{1}{n} \sum_{(s,a,r,s^+) \in \mathcal{D}} (f_j(s,a))^2 = \frac{1}{n} \sum_{i=1}^n (\widehat{u}_j^\top \phi_i)^2 = \widehat{u}_j^\top \widehat{\Sigma} \widehat{u}_j = \widehat{\lambda}_j.$$

 Now, squaring and summing over the constraints and using Eq. (64) yields

$$d\frac{L^2}{n} \gtrsim \sum_{j=1}^m \langle \frac{\widehat{u}_j^\top \phi}{\sqrt{\widehat{\lambda}_j + \lambda}}, \phi^\top(w - \mathcal{P}^\pi(w)) \rangle_\mu^2$$

$$= \sum_{j=1}^m \left[ \frac{\widehat{u}_j^\top}{\sqrt{\widehat{\lambda}_j + \lambda}} \mathbb{E}_\mu \phi \phi^\top (w - \mathcal{P}^\pi(w)) \right]^2$$

$$= \sum_{j=1}^m \left[ \frac{\widehat{u}_j^\top}{\sqrt{\widehat{\lambda}_j + \lambda}} \underbrace{\Sigma(w - \mathcal{P}^\pi(w))}_{\overset{def}{=} y} \right]^2$$

$$= y^\top \left( \sum_{j=1}^m \frac{\widehat{u}_j \widehat{u}_j^\top}{\widehat{\lambda}_j + \lambda} \right) y$$

$$= y^\top \left( \widehat{\Sigma} + \lambda I \right)^{-1} y$$

$$\gtrsim y^\top \Sigma_\lambda^{-1} y.$$

 The last inequality holds via Lemma 14 *(Covariance Concentration)* with probability at least $1 - \delta$
 since $\lambda$ is a large enough regularizer. Let us complete the quadratic form:

$$\|y + \lambda(w - \mathcal{P}^\pi(w))\|_{\Sigma_\lambda^{-1}}^2 \leq (\|y\|_{\Sigma_\lambda^{-1}} + \lambda\|(w - \mathcal{P}^\pi(w))\|_{\Sigma_\lambda^{-1}})^2 \lesssim \|y\|_{\Sigma_\lambda^{-1}}^2 + \lambda.$$

 Therefore, adding $\lambda$ to both sides of the prior display and noticing that $\lambda \lesssim \frac{L^2}{n}$ gives

$$d\frac{L^2}{n} \gtrsim \|y + \lambda(w - \mathcal{P}^\pi(w))\|_{\Sigma_\lambda^{-1}}^2$$

$$= (w - \mathcal{P}^\pi(w)) \Sigma_\lambda^\top \left( \Sigma_\lambda^{-1} \right) \Sigma_\lambda (w - \mathcal{P}^\pi(w))$$

$$= (w - \mathcal{P}^\pi(w))(\Sigma_\lambda)(w - \mathcal{P}^\pi(w))$$

$$= \|(w - \mathcal{P}^\pi(w))\|_{\Sigma_\lambda}^2.$$

 # F Proof of Theorem 2

1335 In this section, we prove the guarantee on our actor-critic procedure stated in Theorem 2.

## F.1 Adversarial MDPs

1337 We now introduce sequence of adversarial MDPs $\{\mathcal{M}_t\}_{t=1}^{T}$ used in the analysis. Each MDP $\mathcal{M}_t$
1338 is defined by the same state-action space and transition law as the original MDP $\mathcal{M}$, but with the
1339 reward functions $R$ perturbed by $R_t$—that is

$$\mathcal{M}_t \stackrel{def}{=} \langle \mathcal{S}, \mathcal{A}, R + R_t, \mathbb{P}, \gamma \rangle. \tag{65}$$

1340 For an arbitrary policy $\pi$, we denote with $Q_t^{\pi}$ and with $A_t^{\pi}$ the action value function and the advantage
1341 function on $\mathcal{M}_t$; the value of $\pi$ from the starting distribution $\nu_{\text{start}}$ is denoted by $V_t^{\pi}$. We immediately
1342 have the following expression for the value function, which follows because the dynamics of $\mathcal{M}_t$ and
1343 $\mathcal{M}$ are identical and the reward function of $\mathcal{M}_t$ equals that of $\mathcal{M}$ plus $R_t$

$$V_t^{\pi} \stackrel{def}{=} \frac{1}{1-\gamma} \mathbb{E}_{\pi} \Big[ R + R_t \Big]. \tag{66}$$

1344 Consider the action value function $\widehat{\underline{Q}}_{\pi_t}$ returned by the critic, and let the reward perturbation
1345 $R_t = \mathcal{B}^{\pi_t} \widehat{\underline{Q}}_{\pi_t}$ be the Bellman error of the critic value function $\widehat{\underline{Q}}_{\pi_t}$. The special property of $\mathcal{M}_t$ is
1346 that the action value function of $\pi_t$ on $\mathcal{M}_t$ equals the critic lower estimate $\widehat{\underline{Q}}_{\pi_t}$.

1347 **Lemma 17** (Adversarial MDP Equivalence). *Given the perturbed MDP $\mathcal{M}_t$ from equation* (65) *with*
1348 $R_t \stackrel{def}{=} \mathcal{B}^{\pi_t} \widehat{\underline{Q}}_{\pi_t}$, *we have the equivalence*

$$Q_t^{\pi_t} = \widehat{\underline{Q}}_{\pi_t}.$$

1349 *Proof.* We need to check that $\widehat{\underline{Q}}_{\pi_t}$ solves the Bellman evaluation equations for the adversarial MDP,
1350 ensuring that $\widehat{\underline{Q}}_{\pi_t}$ is the action-value function of $\pi_t$ on $\mathcal{M}_t$. Let $\mathcal{T}_t^{\pi_t}$ be the Bellman evaluation
1351 operator on $\mathcal{M}_t$ for policy $\pi_t$. We have

$$\widehat{\underline{Q}}_{\pi_t} - \mathcal{T}_t^{\pi_t}(\widehat{\underline{Q}}_{\pi_t}) = \widehat{\underline{Q}}_{\pi_t} - \mathcal{T}^{\pi_t}(\widehat{\underline{Q}}_{\pi_t}) - R_t = \mathcal{B}^{\pi_t}\widehat{\underline{Q}}_{\pi_t} - \mathcal{B}^{\pi_t}\widehat{\underline{Q}}_{\pi_t} = 0.$$

1352 Thus, the function $\widehat{\underline{Q}}_{\pi_t}$ is the action value function of $\pi_t$ on $\mathcal{M}_t$, and it is by definition denoted by
1353 $Q_t^{\pi_t}$. $\square$

1354 This lemma shows that the action-value function $\widehat{\underline{Q}}_{\pi_t}$ computed by the critic is equivalent to the
1355 action-value function of $\pi_t$ on $\mathcal{M}_t$. Thus, we can interpret the critic as performing a model-based
1356 pessimistic estimate of $\pi_t$; this view is useful in the rest of the analysis.

## F.2 Equivalence of Updates

1358 The second step is to establish the equivalence between the update rule (22), or equivalently as the
1359 update (67a), to the exponentiated gradient update rule (67b).

1360 **Lemma 18** (Equivalence of Updates). *For linear Q-functions of the form $Q_t(s, a) = \langle w_t, \phi(s, a) \rangle$,*
1361 *the parameter update*

$$\pi_{t+1}(a \mid s) \propto \exp(\phi(s, a)^{\top}(\theta_t + \eta w_t)), \tag{67a}$$

1362 *is equivalent to the policy update*

$$\pi_{t+1}(a \mid s) \propto \pi_t(a \mid s) \exp(\eta Q_t(s, a)), \qquad \pi_1(a \mid s) = \frac{1}{|\mathcal{A}_s|}. \tag{67b}$$

*Proof.* We prove this claim via induction on $t$. The base case ($t = 1$) holds by a direct calculation. Now let us show that the two update rules update $\pi_t$ in the same way. As an inductive step, assume that both rules maintain the same policy $\pi_t \propto \exp(\phi(s,a)^\top \theta_t)$ at iteration $t$; we will show the policies are still the same at iteration $t + 1$. At any $(s, a)$, we have

$$\pi_{t+1}(a \mid s) \propto \exp(\phi(s,a)^\top(\theta_t + \eta w_t)) \propto \exp(\phi(s,a)^\top \theta_t) \exp(\eta \phi(s,a)^\top w_t)$$
$$\propto \pi_t(a \mid s) \exp(\eta Q_t(s,a)).$$

$\square$

Recall that $\theta_t$ is the parameter associated to $\pi_t$ and that $w_t$ is the parameter associated to $\widehat{\underline{Q}}_{\pi_t}$. Using Lemma 18 together with Lemma 17 we obtain that the actor policy $\pi_t$ satisfies through its parameter $\theta_t$ the mirror descent update rule (67b) with $Q_t = \widehat{\underline{Q}}_{\pi_t} = Q_t^{\pi_t}$ and $\pi_1(a \mid s) = 1/|\mathcal{A}_s|$, $\forall(s, a)$. In words, the actor is using Mirror descent to find the best policy on the sequence of adversarial MDPs $\{\mathcal{M}_t\}$ implicitly identified by the critic.

### F.3 Mirror Descent on Adversarial MDPs

Our third step is to analyze the behavior of mirror descent on the MDP sequence $\{\mathcal{M}_t\}_{t=1}^T$, and then translate such guarantees back to the original MDP $\mathcal{M}$. The following result provides a bound on the average of the value functions $\{V^{\pi_t}\}_{t=1}^T$ induced by the actor's policy sequence. This bound involves a form of optimization error[8] given by

$$\mathcal{E}_{opt}(T) = 2\sqrt{\frac{2 \log |\mathcal{A}|}{T}},$$

as is standard in mirror descent schemes. It also involves the *perturbed rewards* given by $R_t \overset{def}{=} \mathcal{B}^{\pi_t} Q_t^{\pi_t}$.

**Lemma 19** (Mirror Descent on Adversarial MDPs)**.** *For any positive integer $T$, applying the update rule* (67b) *with $Q_t = Q_t^{\pi_t}$ for $T$ rounds yields a sequence such that*

$$\frac{1}{T}\sum_{t=1}^{T}\left[V^{\widetilde{\pi}} - V^{\pi_t}\right] \le \frac{1}{1-\gamma}\left\{\mathcal{E}_{opt}(T) + \frac{1}{T}\sum_{t=1}^{T}\left[-\mathbb{E}_{\widetilde{\pi}}R_t + \mathbb{E}_{\pi_t}R_t\right]\right\}, \qquad (68)$$

*valid for any comparator policy $\widetilde{\pi}$.*

See Appendix F.6 for the proof.

To be clear, the comparator policy $\widetilde{\pi}$ need belong to the soft-max policy class. Apart from the optimization error term, our bound (68) involves the behavior of the perturbed rewards $R_t$ along the comparator $\widetilde{\pi}$ and $\pi_t$, respectively. These correction terms arise because the actor performs the policy update using the action-value function $Q_t^{\pi_t}$ on the perturbed MDPs instead of the real underlying MDP.

### F.4 Pessimism: Bound on $\mathbb{E}_{\pi_t} R_t$

The fourth step of the proof is to leverage the pessimistic estimates returned by critic to simplify equation (68). Using Lemma 9 and the definition of adversarial reward $R_t$ we can write

$$\widehat{V}_{\min}^{\pi} - V^{\pi_t} = \frac{1}{1-\gamma}\langle \mathbb{1}, \mathcal{B}^{\pi_t}\widehat{\underline{Q}}_{\pi_t}\rangle_{\pi_t} = \frac{1}{1-\gamma}\mathbb{E}_{\pi_t}\mathcal{B}^{\pi_t}\widehat{\underline{Q}}_{\pi_t} = \frac{1}{1-\gamma}\mathbb{E}_{\pi_t}R_t.$$

Since weak realizability holds, Theorem 3 guarantees that $\widehat{V}_{\min}^{\pi} \le V^{\pi}$ uniformly for all $\pi \in \Pi$ with probability at least $1 - \delta$. Coupled with the prior display, we find that

$$\mathbb{E}_{\pi_t}R_t \le 0. \qquad (69)$$

Using the above display, the result in Eq. (68) can be further upper bounded and simplified.

---

[8]Technically, this error should depend on $|\mathcal{A}_s|$, if we were to allow the action spaces to have varyign cardinality, but we elide this distinction here.

 **F.5 Concentrability: Bound on $\mathbb{E}_{\widetilde{\pi}} R_t$**

1397 The term $\mathbb{E}_{\widetilde{\pi}} R_t$ can be interpreted as an approximate concentrability factor for the approximate
1398 algorithm that we are investigating.

1399 **Bound under only weak realizability:** Lemma 15 gives with probability at least $1 - \delta$ that any
1400 surviving $Q$ in $\mathcal{C}_n^{\pi_t}$ must satisfy: $\|w - w_\star^{\pi_t}\|_{\Sigma_{\lambda,\mathrm{Boot}}^{+\pi_t}}^2 \lesssim \frac{d\rho}{n}$ where $w_\star^{\pi_t}$ is the parameter associated to
1401 the weak solution $Q_\star^{\pi_t}$. Such bound must apply to the parameter $w_t \in \widehat{\mathcal{C}}_n^{\pi_t}$ identified by the critic.[9]

1402 We are now ready to bound the remaining adversarial reward along the distribution of the comparator
1403 $\widetilde{\pi}$.

$$
\begin{aligned}
|\mathbb{E}_{\widetilde{\pi}} R_t| &= |\mathbb{E}_{\widetilde{\pi}} \mathcal{B}^{\pi_t} \widehat{\underline{Q}}_{\pi_t}| \\
&\overset{(i)}{=} |\mathbb{E}_{\widetilde{\pi}} (\phi - \gamma \phi^{+\pi_t})^\top (w_t - w_\star^{\pi_t})| \\
&\leq \|\mathbb{E}_{\widetilde{\pi}} [\phi - \gamma \phi^{+\pi_t}]\|_{(\Sigma_{\lambda,\mathrm{Boot}}^{+\pi_t})^{-1}} \|w_t - w_\star^{\pi_t}\|_{\Sigma_{\lambda,\mathrm{Boot}}^{+\pi_t}} \\
&\leq c \sqrt{\frac{d\rho}{n}} \sup_{\pi \in \Pi} \left\{ \|\mathbb{E}_{\widetilde{\pi}} [\phi - \gamma \phi^{+\pi}]\|_{(\Sigma_{\lambda,\mathrm{Boot}}^{+\pi})^{-1}} \right\}.
\end{aligned}
\tag{70}
$$

1404 Step (i) follows from the expression (63) for the weak Bellman error, along with the definition of the
1405 weak solution $Q_\star^{\pi_t}$.

1406 **Bound under weak Bellman closure:** When Bellman closure holds we proceed analogously. The
1407 bound in Lemma 16 ensures with probability at least $1 - \delta$ that $\|w - \mathcal{P}^{\pi_t}(w)\|_{\Sigma_\lambda}^2 \leq c \frac{d\rho}{n}$ for all
1408 $w \in \mathcal{C}_n^{\pi_t}$; as before, this relation must apply to the parameter chosen by the critic $w_t \in \widehat{\mathcal{C}}_n^{\pi_t}$. The
1409 bound on the adversarial reward along the distribution of the comparator $\widetilde{\pi}$ now reads

$$
\begin{aligned}
|\mathbb{E}_{\widetilde{\pi}} R_t| = |\mathbb{E}_{\widetilde{\pi}} \mathcal{B}^{\pi_t} \widehat{\underline{Q}}_{\pi_t}| &\overset{(i)}{=} |\mathbb{E}_{\widetilde{\pi}} \phi^\top (w_t - \mathcal{P}^{\pi_t}(w_t))| \\
&\leq \|\mathbb{E}_{\widetilde{\pi}} \phi\|_{\Sigma_\lambda^{-1}} \|w_t - \mathcal{P}^{\pi_t}(w_t)\|_{\Sigma_\lambda} \\
&\leq c \|\mathbb{E}_{\widetilde{\pi}} \phi\|_{\Sigma_\lambda^{-1}} \sqrt{\frac{d\rho}{n}}.
\end{aligned}
\tag{71}
$$

1410 Here step (i) follows from the expression (64) for the Bellman error under weak closure.

1411 **F.6 Proof of Lemma 19**

1412 We now prove our guarantee for a mirror descent procedure on the sequence of adversarial MDPs.
1413 Our analysis makes use of a standard result on online mirror descent for linear functions (e.g., see
1414 Section 5.4.2 of Hazan [Haz21]), which we state here for reference. Given a finite cardinality set
1415 $\mathcal{X}$, a function $f : \mathcal{X} \to \mathbb{R}$, and a distribution $\nu$ over $\mathcal{X}$, we define $f(\nu) \overset{def}{=} \sum_{x \in \mathcal{X}} \nu(x) f(x)$. The
1416 following result gives a guarantee that holds uniformly for any sequence of functions $\{f_t\}_{t=1}^T$, thereby
1417 allowing for the possibility of adversarial behavior.

1418 **Proposition 5** (Adversarial Guarantees for Mirror Descent). *Suppose that we initialize with the*
1419 *uniform distribution $\nu_1(x) = \frac{1}{|\mathcal{X}|}$ for all $x \in \mathcal{X}$, and then perform $T$ rounds of the update*

$$
\nu_{t+1}(x) \propto \nu_t(x) \exp(\eta f_t(x)), \quad \text{for all } x \in \mathcal{X},
\tag{72}
$$

1420 *using $\eta = \sqrt{\frac{\log|\mathcal{X}|}{2T}}$. If $\|f_t\|_\infty \leq 1$ for all $t \in [T]$ then we have the bound*

$$
\frac{1}{T} \sum_{t=1}^T \left[ f_t(\widetilde{\nu}) - f_t(\nu_t) \right] \leq \mathcal{E}_{opt}(T) \overset{def}{=} 2\sqrt{\frac{2\log|\mathcal{X}|}{T}}.
\tag{73}
$$

1421 *where $\widetilde{\nu}$ is any comparator distribution over $\mathcal{X}$.*

---

[9] We abuse the notation and write $w \in \widehat{\mathcal{C}}_n^\pi$ in place of $Q \in \widehat{\mathcal{C}}_n^\pi$

1422  We now use this result to prove our claim. So as to streamline the presentation, it is convenient to
1423  introduce the advantage function corresponding to $\pi_t$. It is a function of the state-action pair $(s, a)$
1424  given by

$$A_t^{\pi_t}(s, a) \overset{def}{=} Q_t^{\pi_t}(s, a) - \mathbb{E}_{a^+ \sim \pi_t(\cdot|s)} Q_t^{\pi_t}(s, a^+).$$

1425  In the sequel, we omit dependence on $(s, a)$ when referring to this function, consistent with the rest
1426  of the paper.

1427  From our earlier observation (66), recall that the reward function of the perturbed MDP $\mathcal{M}_t$ corre-
1428  sponds to that of $\mathcal{M}$ plus the perturbation $R_t$. Combining this fact with a standard simulation lemma
1429  (e.g., [K$^+$03]) applied to $\mathcal{M}_t$, we find that

$$V^{\widetilde{\pi}} - V^{\pi_t} = V_t^{\widetilde{\pi}} - V_t^{\pi_t} + \frac{1}{1-\gamma}\Big[ - \mathbb{E}_{\widetilde{\pi}} R_t + \mathbb{E}_{\pi_t} R_t \Big] = \frac{1}{1-\gamma}\Big[ \mathbb{E}_{\widetilde{\pi}} A_t^{\pi_t} - \mathbb{E}_{\widetilde{\pi}} R_t + \mathbb{E}_{\pi_t} R_t \Big].$$
(74a)

1430  Now for any given state $s$, we introduce the linear objective function

$$f_t(\nu) \overset{def}{=} \mathbb{E}_{a \sim \nu} Q_t^{\pi_t}(s, a) = \sum_{a \in \mathcal{A}} \nu(a) Q_t^{\pi_t}(s, a),$$

1431  where $\nu$ is a distribution over the action space. With this choice, we have the equivalence

$$\mathbb{E}_{a \sim \widetilde{\pi}} A_t^{\pi_t}(s, a) = f_t\big(\widetilde{\pi}(\cdot \mid s)\big) - f_t\big(\pi_t(\cdot \mid s)\big),$$

1432  where the reader should recall that we have fixed an arbitrary state $s$. Consequently, applying the
1433  bound (73) with $\mathcal{X} = \mathcal{A}$ and these choices of linear functions, we conclude that

$$\frac{1}{T} \sum_{t=1}^{T} \mathbb{E}_{a \sim \widetilde{\pi}} A_t^{\pi_t}(s, a) \le \mathcal{E}_{opt}(T).$$
(74b)

1434  This bound holds for any state, and also for any average over the states.

1435  We now combine the pieces to conclude. By computing the average of the bound (74a) over all $T$
1436  iterations, we find that

$$\frac{1}{T} \sum_{t=1}^{T} \Big[ V^{\widetilde{\pi}} - V^{\pi_t} \Big] \le \frac{1}{1-\gamma} \left\{ \frac{1}{T} \sum_{t=1}^{T} \mathbb{E}_{\widetilde{\pi}} A_t^{\pi_t} + \frac{1}{T} \sum_{t=1}^{T} \Big[ - \mathbb{E}_{\widetilde{\pi}} R_t + \mathbb{E}_{\pi_t} R_t \Big] \right\}$$

$$\le \frac{1}{1-\gamma} \left\{ \mathcal{E}_{opt}(T) + \frac{1}{T} \sum_{t=1}^{T} \Big[ - \mathbb{E}_{\widetilde{\pi}} R_t + \mathbb{E}_{\pi_t} R_t \Big] \right\},$$

1437  where the final inequality follow from the bound (73), applied for each $s$. We have thus established
1438  the claim.