# OpenReview forum: "Bellman Residual Orthogonalization for Offline Reinforcement Learning"
_NeurIPS.cc/2022/Conference — NeurIPS 2022 Accept_

### Official Review · Reviewer_xTJV · 2022-07-09

**Rating:** 7
**Confidence:** 4
**Soundness:** 4 excellent
**Presentation:** 2 fair
**Contribution:** 3 good

**Summary:**

This work studies offline policy evaluation and offline policy optimization. The setting is that of discounted rewards, model-free offline RL with function approximation. The main contribution of the paper is a general framework for studying offline policy evaluation through the lens of weak convergence of the Bellman error to zero with respect to a family of test functions. This allows for stating very general policy evaluation and optimization bounds under somewhat weak but reasonable conditions (Assumption 1). The authors give meaningful interpretations of their bounds under stronger assumptions like Bellman closure or using importance sampling weights. Further, the authors recover some versions of some prior work results in the setting of linear MDPs.

**Questions:**

Is weak realizability indeed a weaker condition than realizability? It seems to me that it would amount to realizability if the test function class for example contained all indicators of (s,a)-pairs.

What is the measure \mu? It seems like throughout the paper it is used as the measure from which the data is collected but I never saw a formal definition outside of using it as the measure under which Assumption 1 holds.

How does the second equality hold in Equation 10? It seems like the measure d^\pi has been swapped for \mu, but this does not seem trivial to me.

Where is b_\pi defined in Proposition 1 and what is it?

What is the definition of a prediction P^\pi(Q) in Definition 1 on page 19? Is it just a mapping from the Q-fucntion class to itself?

What is K^\pi_* on line 750?

In Lemma 6 it seems like \lambda can become arbitrarily small so is there no explicit dependence on the number of mixture components?

Where is Y_n(Q,\pi) defined (display under line 1172)?


**Limitations:**

This work is theoretical and it is hard to judge what the societal impact will be.

**Strengths And Weaknesses:**

Strengths:

1.) Originality -- the idea of relaxing some of the known conditions under which offline learning is possible to their weak counterparts is novel and the main theoretical result is novel to the best of my understanding.

2.) Quality -- the generality of Theorem 3 is very nice.

3.) Clarity and significance -- showing how to use the derived results in the linear setting exhibits potentially a general recipe for deriving meaningful bounds in the function approximation setting with weaker conditions than in prior work.

Weaknesses:

1.) Computational efficiency -- the key results make use of computing upper and lower empirical value functions for a given policy \pi, which requires optimizing over the empirical constraint set depending on the TD error and the Q-function class. While for linear MDPs this can be done some-what efficiently, in general it seems like such a computation would require time which is linear in the size of the test function class. Given that in two of the demonstrated examples (LSTD error test space and Bellman test space) the test function class has complexity proportional to the complexity of the Q-function class it is unclear how to efficiently carry out such computations beyond linear models.

2.) Presentation -- the paper is somewhat technical and the main results require introducing a good amount of notation. This, however, is not done in a very good way. I found myself often not being able to recall or understand notation introduced earlier in the paper. It does not help that a lot of the notation is also introduced on the fly and scattered between different sections. Further, given that the main result is bounding the off-policy width through OPC, it would have been good to give interpretable bounds soon after the OPC is introduced in Eq. 10. Some of these bounds are hidden in Appendix A, e.g. Lemma 4, Lemma 6, and it might have suited the presentation better if they were exhibited in the main paper.

Additional comments:

-- the notation used in the proof in C.1.1 for C^\pi_n is not good. It is important to include the dependence on \rho, otherwise the first inequality in the display under line 1081 would not hold.

-- the proof of Theorem 3 seems to be structured around using Proposition 4 and checking that Equation 37 holds, however, this is not done explicitly or at least I have missed it. Otherwise, Proposition 4 does not seem to be used anywhere in the paper.

Overall I think this is a good paper, however, improving the presentation and discussing computationally efficient algorithms beyond the linear setting can strengthen the paper even further.

---

> ### Author Response · Authors · 2022-08-02
> **Rebuttal to xTJV**
>
> We thank the reviewer for the very detailed feedback and for spotting clarity issues and typos. We also welcome the comment on the writing style: as the reviewer notices, in this work we introduced the notation in different places to avoid overwhelming the reader at the beginning, but we agree that this bring other trade-offs. We will try to improve this in the final draft. In addition, we will give interpretable bounds soon after the OPC is introduced in the long version of the paper (here they are moved to the appendix due to space reason).
>
> We will also address the reviewer's 'Additional comments', while below we focus on the reviewer’s main questions.
>
> '*Is weak realizability indeed a weaker condition than realizability? It seems to me that it would amount to realizability if the test function class for example contained all indicators of (s,a)-pairs.*'
>
> The reviewer is correct. But in practice, if the state-action space is large, we use a smaller test class. The idea about weak realizability is that the algorithm would not penalize predictors that are point-wise incorrect as long as they are correct on average.
>
> '*What is the measure $\mu$?*' It is the measure that generates the state-action pairs in the dataset. It is implicitly defined around assumption 2, with a relaxed definition given in appendix. However, as the reviewer notices, $\mu$ is used before reaching that part of the paper, and therefore it needs to be defined upfront. We will define it in section 2 “Background and set-up”.
>
> '*How does the second equality hold in Equation 10? It seems like the measure $d^\pi$ has been swapped for $\mu$, but this does not seem trivial to me.*'
>
> Indeed there is a typo - on the right hand side $\pi$ must be used in the inner product instead of $\mu$
>
> '*Where is $b_\pi$ defined in Proposition 1 and what is it?*'
> $b_\pi$ is a normalization constant to ensure the test function is bounded between [0,1].
> Concretely, one can take $b_\pi = \sup_{s,a} \frac{d^\pi(s,a)}{\mu(s,a)}$.
>
> '*What is the definition of a prediction $P^\pi(Q)$ in Definition 1 on page 19? Is it just a mapping from the Q-fucntion class to itself?*'
>
> Yes, we could have written `$\tilde Q$ such that eq (28) is satisfied’. Here $P^\pi(Q)$ highlights that this can be interpreted as an operator from the space $\mathcal Q$ onto itself.
>
> '*What is $K^\pi_\star$ on line 750?*'
> It is defined right above in (29); we will add the reference inline as it is indeed easy to miss.
>
> '*In Lemma 6 it seems like \lambda can become arbitrarily small so is there no explicit dependence on the number of mixture components?*'
>
> $\lambda$ does need to satisfy a lower bound for the theorems (and concentrability coefficients) to hold, see line 194 in Thm 1 (there we write it as an equality, but it is really a lower bound on the value of lambda).
>
> '*Where is $Y_n(Q,\pi)$ defined (display under line 1172)?*'
>
> This is a typo, and it should ready $Z_n$ instead of $Y_n$.

---

### Official Review · Reviewer_HGjj · 2022-07-13

**Rating:** 7
**Confidence:** 3
**Soundness:** 3 good
**Presentation:** 2 fair
**Contribution:** 3 good

**Summary:**

The authors propose an offline policy evaluation and optimization method introducing a test class (discriminator). The policy optimization method is performed pessimistically by constructing the version space. In the linear case, it is at least computationally efficient. They also propose a new off-policy coefficient, which would result in several known or new coefficients in various scenarios.


**Questions:**

Can we consider an NPG-type policy optimization method with general function approximation with this framework like [XCJ+21] beyond the linear case? If so, that's interesting since their result relies on Bellman closeness.

**Ethics Review Area:**

["I don’t know"]

**Limitations:**

Not applied to this work.

**Strengths And Weaknesses:**

I think it is a thought-provoking paper. I will vote for acceptance. But I think some parts of the acknowledgments are not exactly fair as I will mention later. It might be better to be fixed.

(Strengths)

* They propose a unified framework with CPC that can capture known results and new results.

* Results in Section 4.3 (a generalization of LSTD) are very interesting.

* In Section 5.1, I feel prop 2 and prop 3 are kind of known. But, viewing it in a unified way is still interesting.  Related to this point, the computationally efficient algorithm (Section 5.2) without Bellman completeness is interesting. (Though I feel this algorithm is known and the bellman completeness type result is known as author acknowledged)

(Weakness)

The downside is some of the concrete results, which come from a unified framework are known. Some of them are well-acknowledged. But I feel some of them are not acknowledged in direct ways. ( Irrespective of that, unified thinking itself is a valuable and thought-provoking contribution! )

More specifically,  for domain-specific results,

* Page 2 bullet point 1 : contribution 1, “however, here we only require concentrability with respect to a comparator policy instead of over all policies in the class.” Umm. I think this is over claimed. It is explicitly mentioned in [JH20]. Though they did not give finite sample results, I feel this part is not so difficult ( I remember the authors mentioned it either). So, I think it is fair to acknowledge like "this observation is obtained in JH20, and we make it formal".  (I see a related acknowledgment is written in a footnote. But this writing way is a bit vague.)

* Page 2 bullet point 4: “Finally, our procedure inherits some form of “multiple robustness”. For example, the two test classes corresponding to Bellman completeness and marginalized importance weights can be used together, and guarantees will be obtained if either Bellman completeness holds or the importance weights are correct” This is also kind of known I feel. In OPE context, e.g., UIJ+21.

---

> ### Author Response · Authors · 2022-08-02
> **Rebuttal to HGjj**
>
> We thank the reviewer for the review and for pointing out that the relation with the literature / acknowledgment of prior results should be improved; we will address this in the updated version. We address the reviewer's only question below.
>
> '*Can we consider an NPG-type policy optimization method with general function approximation with this framework like [XCJ+21] beyond the linear case? If so, that’s interesting since their result relies on Bellman closeness.*'
>
> Yes - we believe the critic can be swapped as the reviewer suggests to relax the need for completeness.

---

### Official Review · Reviewer_eZ2Z · 2022-07-15

**Rating:** 8
**Confidence:** 3
**Soundness:** 4 excellent
**Presentation:** 3 good
**Contribution:** 4 excellent

**Summary:**

This paper studied and analyzed a principle way to approximate bellman equations, where the authors constrain the value function in a set that was controlled by user-defined test functions and empirical Bellman errors. Under this unified principle, the authors are able to construct confidence intervals for off-policy evaluation, and also performing pessimistic offline policy learning using the lower bound. Besides, the authors also establish connections to concentrability coefficients studied in past work

**Questions:**

- [4] is a recent / concurrent work, whose main objective is started with density ratio. I wonder if it is possible to consider the Q as the test function, and construct empirical set for the density ratio function? And what's the connection of this new principle and the principles discussed in the paper( density ratio as the test function) ?

- Would you mind explaining the notion of $o$? What would it could be in the real world? like context?



[1] Jiang, et.al . Minimax value interval for off-policy evaluation and policy optimization, NeurIPS 2020.
[2] Feng, et.al. Accountable Off-Policy Evaluation With Kernel Bellman Statistics, ICML 2020.
[3] Feng, et.al. Non-asymptotic Confidence Intervals of Off-policy Evaluation: Primal and Dual Bounds, ICLR 2021.
[4] Zhan, et.al. Offline Reinforcement Learning with Realizability and Single-policy Concentrability, COLT 2022.


**Limitations:**

There is not potential negative societal impact of this work.

**Strengths And Weaknesses:**

- **Originality**.
The idea of constructing the empirical set and then use the empirical set to construct confidence interval has been explored in the prior works [1,2,3]. For instance, [2,3] aim to provide computational confidence interval, where the authors constrain the test function into a rkhs space, and the empirical set in [3] seems to be the same as Equation (7) in this paper.  **But**, [2,3] mainly focus on how to obtain computational confidence intervals, this paper provide more theoretical results such as offline policy optimization and results under weaker assumption.  Moreover,  the discussion in Section 4, the connections to concentrability coefficients in the prior works are very interesting.

- **Quality**.
This paper is technically sound. The authors provide many new results under this framework (such as offline policy improvement).

- **Clarity**.
This paper is well-written. I am new to this written style, the readers had better to read the appendix to get the background. Besides, is it possible to have a conclusion section that includes summary of the work and also some discussions on the future work?

- **Significance**.
I think this is a very good paper, especially I like the explanation of connection between the Concentrability Coefficients and test spaces.

---

> ### Author Response · Authors · 2022-08-02
> **Rebuttal to eZ2Z**
>
> We thank the reviewer for the review and for the insightful connections with the literature. We will add a conclusion as suggested by the reviewer (it was omitted for space reason).
> Below we address the reviewer’s main questions.
>
> '*[4] is a recent / concurrent work, whose main objective is started with density ratio. I wonder if it is possible to consider the Q as the test function, and construct empirical set for the density ratio function?* '
>
> Yes. This is a very interesting question, and a possible avenue for future research. As we understand, the reviewer wants to 'swap' the role of the test function and predictors: here the new aim is to find the correct density ratio, and use the Q values to 'test' the density ratios. We did think about this, but we did not investigate it in detail. Indeed, the paper [1] cited by the reviewer already suggests that this is possible  and it will bring different results (e.g., weak realizability would now refer to the density ratios).
>
> '*And what’s the connection of this new principle and the principles discussed in the paper ( density ratio as the test function) ?*'
>
> Indeed, the two should coincide for very specific choices, such as the one suggested by the reviewer.
>
> '*Would you mind explaining the notion of $o$? What would it could be in the real world? like context?*'
>
> In case that the dataset originate from multiple policies, $o$ can be used to 'tag' the specific policy that generated a certain sample.
> We have an example of this in appendix 'A.5.2 Mixture distributions' and 'A.5.3 Bellman Rank for off-policy evaluation'.

---

> > ### Comment · Reviewer_eZ2Z · 2022-08-08
> > **Reponse to the Rebuttal to eZ2Z**
> >
> > Thanks the authors for the detailed answer. I have no further questions regarding the paper and personally I think it is a very good paper.
> > I keep my current score.

---

### Official Review · Reviewer_ikh4 · 2022-07-18

**Rating:** 6
**Confidence:** 3
**Soundness:** 2 fair
**Presentation:** 3 good
**Contribution:** 2 fair

**Summary:**

The main idea of Bellman residual orthogonalization proposed by this paper is to approximate the Bellman equations by enforcing their validity along a user-defined space of test functions. Then, this work utilizes this idea to construct confidence intervals for OPE as well as to optimize over policies in the model-free offline reinforcement learning with function approximation setting.

In contrast to the standard importance sampling (IS) and regression based methods for OPE, the approach, derived from the Bellman residual orthogonalization principle, is a kind of weight learning algorithms. Specifically, the proposed method leverages an auxiliary function class to either encode the marginalized importance weights of the target policy or estimates of the Bellman errors. In particular, the authors propose a sample-based approximation of this idea via self-normalization and regularization.



**Questions:**

Firstly, what are the key advantages of the methods for OPE and policy optimization based on Bellman residual orthogonalization over standard approaches for both tasks in the offline setting?

Secondly, given the rationale of the proposed principle, the current version may be further improved by providing analysis and guidance regarding which tasks in RL, besides OPE, could be better solved using the same idea.

Last but not least, test functions play a critical role in the proposed methods and are used to encode domain knowledge of interest. Accordingly, it is very important carefully design the test class. Although the paper points out that the choice of the test functions are contingent on the problems and several principles that the design of the test function space should satisfy, the current version lacks a systemic guidance for doing so.

**Limitations:**

As this work focusing on a RL method from a theoretical aspect, the authors have analyzed certain technical limitation, such as the way to design a good test class, the pros and cons of using a combined test classes, etc.


**Strengths And Weaknesses:**

The proposed Bellman residual orthogonality seeks to control the Bellman error in a weighted-average way where the weights are derived from a set of test functions chosen by the users. Therefore, the choice of the test class is crucial and contingent on the problem of interest, but it is also challenging to choose the right one. This paper has discussed several overarching principles that should be adopted to guide the design of the test function space. For instance, the test class should be delicately designed in order to encode domain knowledge of interest. However, there is a gap between such principles and practical procedures to do so. In this sense, this current version could be improved by analyzing the pros and cons of different test functions in-depth and developing a principled way to make such decision.

In addition, this paper utilizes the aforementioned principle to tackle two RL problems, off-policy evaluation and policy optimization, in the offline RL setting with function approximation. As these two RL tasks have been investigated extensively and many existing approaches are available, it is important to compare the performance of the method developed in this paper compared to existing techniques, which would help verify the technical soundness and practical effectiveness of the proposed idea.

---

> ### Author Response · Authors · 2022-08-02
> **Rebuttal to ikh4**
>
> We thank the reviewer for providing the review. Below we address the reviewer’s questions.
>
> '*Firstly, what are the key advantages of the methods for OPE and policy optimization based on Bellman residual orthogonalization over standard approaches for both tasks in the offline setting?*'
>
> As the reviewer notices, there exist several standard approaches in offline RL. Orthogononalizing the residual has an intuitive appeal, and the idea is flexible enough to recover several prior approaches using the same algorithm simply by varying the test class. However the approach presented here is more general: as the test class can be chosen in several ways, many more RL domains can potentially be learned by different choices of the test functions using the same algorithm.
>
> '*Secondly, given the rationale of the proposed principle, the current version may be further improved by providing analysis and guidance regarding which tasks in RL, besides OPE, could be better solved using the same idea.*'
>
> We agree with the reviewer -- as a start, it would be interesting for example to examine the exploration setting.
>
> '*Last but not least, test functions play a critical role in the proposed methods and are used to encode domain knowledge of interest. Accordingly, it is very important carefully design the test class. Although the paper points out that the choice of the test functions are contingent on the problems and several principles that the design of the test function space should satisfy, the current version lacks a systemic guidance for doing so.*'
>
> As the reviewer notices, although there is a discussion at the beginning of section 4, there is no `silver bullet’ to choose the test functions. Ultimately some domain knowledge and understanding of the specific problem is needed to choose the test class, much like it is needed for the Q-class. However, the examples reported in section 4 and 5, as well as in the appendix, can be taken as a starting point to build some intuition.

---

### Author Response · Authors · 2022-08-02
**Rebuttal**

We thank all reviewers for their valuable and in-depth comments. Below we address the reviewers' main questions.

---

### Meta-Review · Area_Chair_BigN · 2022-08-24

**Recommendation:** Accept
**Confidence:** Certain

**Metareview:**

All reviewers recommend acceptance and the meta-reviewer agrees. The reviewers appreciated the generality and the new point of view introduced by the Bellman residual orthogonalization framework for offline RL. Several existing results and techniques are recovered by this paper which can be viewed critical, in the sense that many results are already known. However, the paper generalizes these results significantly, deriving several new results, e.g. the generalization of LSTD to output confidence intervals, and requiring weaker assumptions to achieve existing results, e.g. computationally tractable policy optimization in the linear setting without Bellman completeness. Unfortunately, it is unclear if and how the analyzed methods can be implemented computationally efficiently beyond the linear case.
Overall, this is a solid technical paper with useful theoretical insights that were appreciated by all reviewers.

**Award:**

No

---

### Decision · Program_Chairs · 2022-09-14

Accept